

# The postcranial anatomy of *Moschorhinus kitchingi* (Therapsida: Therocephalia) from the Karoo Basin of South Africa

Brandon P. Stuart[1,2,3], Adam K. Huttenlocker[4] and Jennifer Botha[2,3,5]

[1] Department of Zoology and Entomology, University of the Free State, Bloemfontein, Free State, South Africa
[2] Evolutionary Studies Institute, University of the Witwatersrand, Johannesburg, Gauteng, South Africa
[3] School of Geosciences, University of the Witwatersrand, Johannesburg, Gauteng, South Africa
[4] Department of Integrative Anatomical Sciences, University of Southern California, Los Angeles, California, United States
[5] University of the Witwatersrand, GENUS: DSI-NRF Centre of Excellence in Palaeosciences, Johannesburg, Gauteng, South Africa

Corresponding author
Brandon P. Stuart,
brandon.stuart8@gmail.com

## ABSTRACT

Therocephalia are an important clade of non-mammalian therapsids that evolved a diverse array of morphotypes and body sizes throughout their evolutionary history. The postcranial anatomy of therocephalians has largely been overlooked, but remains important towards understanding aspects of their palaeobiology and phylogenetic relationships. Here, we provide the first postcranial description of the large akidnognathid eutherocephalian *Moschorhinus kitchingi* by examining multiple specimens from fossil collections in South Africa. We also compare the postcranial anatomy with previously described therocephalian postcranial material and provide an updated literature review to ensure a reliable foundation of comparison for future descriptive work. *Moschorhinus* shares all the postcranial features of eutherocephalians that differentiate them from early-diverging therocephalians, but is differentiated from other eutherocephalian taxa by aspects concerning the scapula, interclavicle, sternum, manus, and femur. The novel anatomical data from this contribution shows that *Moschorhinus* possessed a stocky bauplan with a particularly robust scapula, humerus, and femur. These attributes, coupled with the short and robust skull bearing enlarged conical canines imply that *Moschorhinus* was well equipped to grapple with and subdue prey items. Additionally, the combination of these attributes differ from those of similarly sized coeval gorgonopsians, which would have occupied a similar niche in late Permian ecosystems. Moreover, *Moschorhinus* was the only large carnivore known to have survived the Permo-Triassic mass extinction. Thus, the subtle but important postcranial differences may suggest a type of niche partitioning in the predator guild during the Permo-Triassic mass extinction interval.

# INTRODUCTION

Therocephalians were a widely distributed and ecomorphologically diverse clade of eutheriodont therapsids that lived from the middle Permian to the Middle Triassic (*Abdala, Rubidge & van den Heever, 2008*; *Huttenlocker, Sidor & Angielczyk, 2015*; *Kammerer & Masyutin, 2018a*). The earliest-diverging therocephalians (Lycosuchidae and Scylacosauridae) were large-bodied (skull lengths up to 400 mm) carnivores that functioned as important components of Guadalupian (middle Permian) macropredatory niches (*Abdala, Rubidge & van den Heever, 2008*; *Abdala et al., 2014b*). The speciose subclade, Eutherocephalia, arose during the late Permian (Lopingian) and rapidly diversified into a wide range of body sizes and ecological niches with taxa consisting of mostly small-to-medium-bodied (skull lengths ranging from 100–200 mm) faunivores (*Huttenlocker, 2014*; *Huttenlocker & Smith, 2017*) and mostly small-bodied (skull lengths ≤ 100 mm) generalist and herbivorous taxa during the Early-to-Middle Triassic (*Abdala et al., 2014a*; *Huttenlocker et al., 2022*).

The Akidnognathidae constitute a geographically widespread lineage of small-to-large-bodied (skull lengths ranging from 100–260 mm) carnivorous eutherocephalians that are characterized by the presence of a laterally expanded or fan-shaped anterior margin of the vomer, enlarged anteriorly facing external nares with an enlarged septomaxilla that broadly overlaps the premaxilla, a separate bony housing for the lower canine fossa formed by the maxilla, five upper incisors, and no pterygoid teeth (*Huttenlocker, 2009*; *Huttenlocker, Sidor & Smith, 2011*; *Liu & Abdala, 2017, 2019*). They also may preserve carinae on the distal cutting edges of the postcanines, like the closely-related Chthonosauridae, and have a tendency to reduce the number of postcanine teeth in some later Permian and Triassic forms (*Olivierosuchus*, *Cerdosuchoides*, *Moschorhinus*) (*Huttenlocker, Sidor & Smith, 2011*; *Huttenlocker, Sidor & Angielczyk, 2015*). The earliest-diverging members of the lineage are known from the upper Permian strata of Russia (*Ivakhnenko, 2011*) and China (*Liu & Abdala, 2017, 2019, 2022*) whereas the remainder of the clade is represented in the upper Permian through Lower Triassic sediments of the Karoo Basin in South Africa (*Huttenlocker, Sidor & Smith, 2011*; *Botha-Brink & Modesto, 2011*). Permo-Triassic akidnognathids (*e.g.*, *Moschorhinus*, *Olivierosuchus*, and *Promoschorhynchus*) are among the most abundant therocephalians that have been recovered from the *Daptocephalus* and *Lystrosaurus declivis* assemblage zones (AZ) of the Karoo Basin and represent a significant portion of theriodont diversity during the transition between Permo-Triassic ecosystems amidst the Permian-Triassic mass extinction (PTME) (*Huttenlocker, Sidor & Smith, 2011*; *Botha-Brink, Huttenlocker & Modesto, 2013*; *Viglietti et al., 2021*; *Kammerer et al., 2023*).

*Moschorhinus kitchingi* is the only eutherocephalian known to have occupied a macropredatory role in late Permian ecosystems. This ecological niche was primarily filled by coeval gorgonopsians, however they did not survive the PTME, making *Moschorhinus* the largest known therapsid predator to cross the Permo-Triassic boundary (PTB) (*Huttenlocker & Botha-Brink, 2013*; *Kammerer et al., 2023*). *Moschorhinus* is only known from the Karoo Basin with the first occurrence from the upper *Daptocephalus*

(*Lystrosaurus maccaigi-Moschorhinus* Subzone) AZ of the Balfour Formation where it is abundant and the last occurrence from the lower *Lystrosaurus declivis* AZ of the Palingkloof Member of the Balfour Formation where it is a rarer component (*Huttenlocker & Botha-Brink, 2013*; *Botha-Brink, Huttenlocker & Modesto, 2013*; *Viglietti, 2020*; *Botha & Smith, 2020*). The cranial morphology of *Moschorhinus* is well known and has been described by previous authors (*e.g.*, *Broom, 1920*; *Boonstra, 1934*; *Brink, 1958b*; *Mendrez, 1974a*; *Durand, 1991*), but although represented by numerous specimens, the postcranial morphology has not been described.

Detailed postcranial descriptions of therocephalians are sparse, in contrast to those of the crania, apart from a few noteworthy contributions. Historically, the first therocephalian to undergo any form of postcranial description was *Ictidosuchus primaevus* by *Broom (1900)*. Subsequently, more accounts of therocephalian taxa were produced by multiple authors (*Haughton, 1918*, *1929*; *Watson, 1931*; *Broom, 1932*, *1936*, *1938*, *1948*; *Boonstra, 1934*, *1954*; *von Huene, 1950*; *Brink, 1958b*; *Sigogneau, 1963*; *Cluver, 1969*). *Attridge (1956)* produced one of the first detailed descriptions of therocephalian postcrania using a mostly complete skeleton of the whaitsioid *Mirotenthes digitipes*. *Boonstra (1964)* and *Cys (1967)* provided significant contributions on the postcranial morphology of early-diverging therocephalians by describing various appendicular material and a complete skeleton from the *Tapinocephalus* AZ. *Kemp (1978)* commented on the functional aspects of the therocephalian skeleton by describing the ilium and hind limb of *Regisaurus jacobi* as well as a mostly complete skeleton of a small therocephalian (*Kemp, 1986*), which he referred to a regisaurid, now considered to represent a juvenile specimen of *Scaloposaurus* (*Huttenlocker et al., 2022*). During this time additional accounts of therocephalian postcrania from outside the Karoo Basin were published (*Hou, 1979*; *Colbert & Kitching, 1981*). The last postcranial description of the 20[th] century was published by *King (1996)*, in which she described a small, disarticulated skeleton, which she tentatively attributed to *Bauria cynops*, but is now referred to *Microgomphodon oligocynus* (*Huttenlocker, 2013*, *2014*; *Huttenlocker & Botha-Brink, 2014*; *Abdala et al., 2014a*).

After almost a decade of hiatus of therocephalian postcranial descriptions, *Fourie & Rubidge (2007*, *2009)* and *Fourie (2013)* published the descriptions of *Olivierosuchus* (see *Botha-Brink, Huttenlocker & Modesto, 2013*) for taxonomic re-assignment of BP/1/3973), a scylacosaurid referred tentatively to *Glanosuchus*, and *Ictidosuchoides*. Additional postcranial descriptions of new specimens of *Olivierosuchus* (*Botha-Brink & Modesto, 2011*), *Promoschorhynchus* (*Huttenlocker, Sidor & Smith, 2011*), *Tetracynodon darti* (*Sigurdsen et al., 2012*), *Simorhinella baini* (*Abdala et al., 2014b*), and *Microgomphodon* (*Abdala et al., 2014a*) were published around this time, and although none of these specimens were complete skeletons and the central focus of these publications was not on the postcranial morphology itself, building on the work of *Fourie & Rubidge (2007*, *2009)* and *Fourie (2013)* more comprehensive comparisons of therocephalian postcranial elements could be made. More recently, descriptions of new therocephalian taxa from the Kotelnich locality in Russia (*Kammerer & Masyutin, 2018a*) and the Naobaogou Formation in China (*Liu & Abdala, 2017*, *2019*, *2020*) were published, although these specimens only preserve partial postcranial elements.

Despite these efforts the therocephalian postcranial skeleton remains understudied. For example, approximately only four taxa have received significant postcranial examinations over the past two decades. Previously this was largely due to the scarcity of well-preserved and diagnostic postcranial material. However, collecting efforts over the past two decades have produced numerous skeletons of different taxa. In an effort to provide fresh data on the postcranial morphology of therocephalians and a basis for future descriptions and phylogenetic analyses, we provide the first postcranial description of the akidnognathid *Moschorhinus kitchingi* by examining multiple specimens that span a range of different body sizes and stratigraphic levels.

## MATERIALS AND METHODS

### Material

The following description is based on previously collected and undescribed specimens referred to *Moschorhinus* (Table 1). All of the specimens facilitating the following description include well-preserved cranial material that were previously used in *Huttenlocker & Botha-Brink (2013)* enabling the positive referral of specimens to *Moschorhinus*. All the specimens selected for this description are housed in the Karoo fossil collections of South Africa and were collected from the upper Permian *Daptocephalus* AZ and Lower Triassic *Lystrosaurus declivis* AZ of the Karoo Basin of South Africa. All the material described in this study was personally examined, measured, and photographed by the authors. Measurements of the material were taken using Mitutoyo digimatic calipers and digital photographs were taken using a Canon 800D camera body equipped with 18–55 and 55–250 mm lenses at varying focal lengths.

### Anatomical terminology and conventions

Previous authors have described discrete regions (*i.e.*, cervical, thoracic, and lumbar) of the presacral axial skeleton of therocephalians (*e.g.*, *Watson, 1931*; *Attridge, 1956*; *Brink, 1958b*; *Cys, 1967*; *Cluver, 1969*; *Kemp, 1986*; *Fourie & Rubidge, 2007*, *2009*; *Botha-Brink & Modesto, 2011*; *Sigurdsen et al., 2012*; *Fourie, 2013*). The recent quantitative study of *Jones et al. (2018)* on the evolution of the specialized and regionalized axial skeleton of mammals suggests this only occurred at the crown mammal radiation in the synapsid lineage and that the presacral region of the therapsid skeleton is comprised of cervical and dorsal (anterior and posterior) units. Following this, we therefore opt to make use of cervical, anterior dorsal, mid-dorsal, posterior dorsal, and refer to specific vertebrae and ribs to highlight salient features for the description of the axial material of *Moschorhinus*.

Historically and more recently, previous authors have described the humerus and femur as extending out laterally and parallel to the ground so that they display dorsal, ventral, anterior, and posterior surfaces (*e.g.*, *Haughton, 1929*; *Boonstra, 1964*; *Kemp, 1978*, *1986*; *Fourie & Rubidge, 2007*, *2009*), following the anatomical conventions used for sprawling animals such as non-therapsid synapsids (*e.g.*, *Romer & Price, 1940*; *Romer, 1956*; *Hopson, 2012*). At least some therapsids are thought to have adopted a dual gait stance, with a sprawled forelimb and a semi-erect or erect hind limb in contrast with the sprawled fore- and hind limbs of the more basal non-therapsid synapsids (*Kemp, 1978*; *Blob, 2001*).

**Table 1 List of specimens of *Moschorhinus kitchingi* examined in this study.**

| Specimen | Description | BSL | Locality and assemblage zone |
|---|---|---|---|
| NMQR 3351 | Skull, lower jaw, articulated axial skeleton, clavicles, scapulae, partial procoracoid and coracoid, interclavicle, sternum, humeri, radius, ulna, partial manus, ilia, ischium, femora, tibia, fibula | 240 | Bokpoort, Wepener, Free State (DAZ) |
| NMQR 3939 | Skull, partial lower jaw, partial articulated axial skeleton, partial scapulae, procoracoid, humerus, radius, ulna (cast representing distal humerus, proximal radius, and proximal ulna), manus, partial ilium, ischium, femur, tibiae | 170 | Nooitgedacht 68, Free State (DAZ) |
| CGS GHG299 | Skull, sternum, and other postcranial elements covered with matrix | 200 | Kommandodrift, Eastern Cape (DAZ) |
| BP/1/4227 | Skull, lower jaw, scapula, humerus, radius, ulna, partial manus | 200 | Admiralty Estates, KwaZulu Natal (LAZ) |
| NMQR 3568 | Skull, two isolated vertebrae, pubis | 198 | Skerpioenkraal, Eastern Cape (LAZ) |
| NMQR 48 | Skull, lower jaw, isolated vertebrae, ribs, partial clavicle, partial scapula, partial procoracoid, coracoids, interclavicle, sternum, proximal humerus, cast and photographs of humerus, distal radius, distal ulna, partial manus, ischium, pubis, proximal femur | 163 | Zeekeoigat, Eastern Cape (unknown) |
| SAM-PK-K10698 | Lower jaw, isolated rib, ilium | – | Lucerne 70, Graaff-Reinet, Eastern Cape (LAZ) |

**Notes:**
Specimens are ordered by Assemblage Zone and basal skull length (BSL).
DAZ, *Daptocephalus* assemblage zone; LAZ, *Lystrosaurus declivis* assemblage zone.

Moreover, some therocephalians have been interpreted as having a facultatively erect hind limb posture capable of variable gaits (*Kemp, 1978*). Based on the morphology of the glenoid of *Moschorhinus* we describe the humerus as extending posterolaterally from the glenoid and parallel to the ground so that it presents with a dorsal, ventral, lateral, and medial surface. On the other hand, we interpret the femur as being held more closely to the sagittal plane so that it presents with an anterior, posterior, lateral, and medial surface. These anatomical directions are used in the following descriptions for the forelimb and hind limb, respectively.

## Systematic palaeontology

THERAPSIDA *Broom, 1905*
EUTHERIODONTIA *Hopson & Barghusen, 1986*
THEROCEPHALIA *Broom, 1903*
AKIDNOGNATHIDAE *von Nopsca, 1928*
*MOSCHORHINUS Broom, 1920*
*MOSCHORHINUS KITCHINGI Broom, 1920*

*Tigrisuchus simus Owen, 1876*
'*Scymnosaurus*' *warreni Broom, 1907*
*Moschorhinus warreni* (*Broom, 1932*)
*Moschorhinus minor Broom, 1936*
*Moschorhinus esterhuyseni Broom, 1940*
*Moschorhinus natalensis Brink, 1958a*

**Holotype**—NHMUK R5698, an incomplete skull missing the posterior end.

**Type locality**—NHMUK R5698 was collected by James William Kitching (1896–1953) (uncle of prolific fossil collector James William Kitching (1922–2003)) near the Bethesda Road in Graaff-Reinet, Eastern Cape Province, Republic of South Africa. The original stratigraphic provenance was documented as "*Cistecephalus* zone" of the Karoo Basin.

**Occurrence**—Occurs in the *Lystrosaurus maccaigi-Moschorhinus* Subzone of the *Daptocephalus* AZ (upper Permian) to Lower Triassic *Lystrosaurus declivis* AZ (Changhsingian to Induan stages), Beaufort Group, Karoo Supergroup; known from numerous localities throughout the Eastern Cape, Free State, and KwaZulu-Natal provinces, Karoo Basin, South Africa. *Huttenlocker & Botha-Brink (2013)* showed that the Triassic record of the genus is typified by fewer fossils and generally smaller sizes than in the Permian.

**Diagnosis**—Medium-to-large eutherocephalian (basal skull length up to 260 mm) with: short, broad rostrum; narrow, slit-like pineal foramen; extremely broad, fan-shaped vomer bearing paired tubercles anteroventrally; crista choanalis broad and rounded; pterygoid teeth completely absent (shared with other akidnognathids); dentary deepened with prominent mental protuberance; upper tooth count I5:pC0–1:C1:PC3–4 (as few as two postcanines in some large specimens); spatulate upper incisors with broad, smooth facets on enamel surface (unstriated); first three upper incisors have mesiodistally elongate figure-8 cross-section; enlarged, saber-like upper canines lacking carinae or serrations (rounded in cross-section); fan-shaped ventral end of the clavicle with a scalloped posteromedial margin; small, shield-shaped interclavicle with reduced lateral rami and smooth posterior ramus; large, circular sternum lacking a posterior notch; robust femur with a dorsoventrally long, triangular internal trochanter.

**Remarks**—*Moschorhinus* specimens are locally abundant throughout the Karoo Basin, and are well represented by numerous cranial and postcranial fossils (*Huttenlocker & Botha-Brink, 2013*). *Mendrez (1974a, 1974b)* tentatively proposed synonymy with *Tigrisuchus*, which was based on a weathered snout collected by A. Bain and described by *Owen (1876)*. Previous authors have referred *Tigrisuchus* to Gorgonopsia (*e.g.*, *Sigogneau, 1970*; *van den Heever, 1987*). A formalized synonymy would maintain the prevailing usage of the popular name *Moschorhinus*, and the older name '*Tigrisuchus*,' which has been in disuse for many decades, should be considered a *nomen oblitum* under ICZN Article 23.9. Other species previously attributed to the genus *Moschorhinus* (*M. minor* Broom, *M. esterhuyseni* Broom, and *M. natalensis* Brink) are also all likely junior synonyms of the type species *M. kitchingi*.

## DESCRIPTION

### Vertebrae

The most complete vertebral column is preserved in NMQR 3351, which preserves 34 articulated vertebrae, comprising 27 presacral, three sacral, and four caudal vertebrae. The

vertebral column is separated by two precise transverse breaks so that the atlas-axis complex and third cervical, fourth cervical–11th dorsal, 12th dorsal–fourth caudal are preserved on separate blocks (Fig. 1A). NMQR 3939 includes fragments of cervical vertebrae on the first block, likely elements of the atlas-axis complex, as well as a set of 15 articulated dorsal vertebrae on the second block (Fig. 1B). None of the centra of NMQR 3351 (apart from the atlas-axis complex and third cervical) or NMQR 3939 are exposed.

### Cervical vertebrae

**Atlas-axis complex**—No proatlantes are preserved in NMQR 3351, but have been described in other therocephalians (*Kemp, 1986*; *Botha-Brink & Modesto, 2011*). The atlas-axis complex is well preserved and only missing the left neural arch of the atlas (Fig. 2). The neural arch is approximately L-shaped in dorsal view, but has a slight anterior projection (Figs. 2A and 2B). The dorsal lamina extends posteriorly to form the postzygapophysis, which articulates with the dorsal surface of the prezygapophysis of the axis (Figs. 2A and 2B). The anterior lamina extends laterally forming the atlantal transverse process (Figs. 2A, 2B, 2E and 2F). The atlantal transverse process is mediolaterally short and extends ventrolaterally from the neural arch (Figs. 2A, 2B, 2E and 2F). Immediately posterior to the distal end of the transverse process, separated by a break, a mediolaterally thin sheet of bone is present that extends posteriorly and is interpreted as the corresponding atlantal rib (Figs. 2I and 2J).

The atlas centrum is preserved directly below the neural arch and the majority of its structure is visible (Figs. 2A, 2B, 2G, and 2H). The dorsal surface is concave, bearing a mediolaterally broad midline trough, which forms the ventral surface of the neural canal (Figs. 2A–2D). Only the left margin of the trough is visible; it is delimited by a sharp ridge, which extends more posteriorly than laterally. The anterior surface of the atlantal centrum is complex and is composed of three articular facets similar to those described for *Scaloposaurus* (*Kemp, 1986*). The three articulatory facets all converge at the centre of the anterior surface where a circular depression is present (Figs. 2E and 2F), which differs from the condition in *Scaloposaurus* where a swelling is present. A dorsolaterally, and slightly anteriorly facing facet is present on either side (the right being covered by the neural arch) for the articulation with the atlantal neural arches (Figs. 2E and 2F). The left facet is relatively mediolaterally broad, with a weakly depressed lateral surface, and a convex posterior margin. Ventral to this facet the surface slopes posteromedially and a deep notch is formed at the most anterior end. The third articulatory facet, for the reception of the atlantal intercentrum, is present ventrally (Figs. 2E and 2F). The ventral margin is convex and is in direct contact with the atlantal intercentrum.

The atlantal intercentrum is a bulbous structure and is approximately kidney shaped in anterior view (Figs. 2E and 2F). The anterolateral margins are convex and the ventral margin is straight. The parapophysis is positioned on the lateral surface in the form of a raised edge to receive the capitulum of the dichocephalous atlantal rib (Figs. 2E and 2F). The tuberculum is still articulated to the distal end of the transverse process of the neural arch (Figs. 2E, 2F, 2I, and 2J). The shaft of the atlantal rib is mediolaterally narrow and dorsoventrally tall, but it is slightly deformed and it is difficult to interpret its true length.

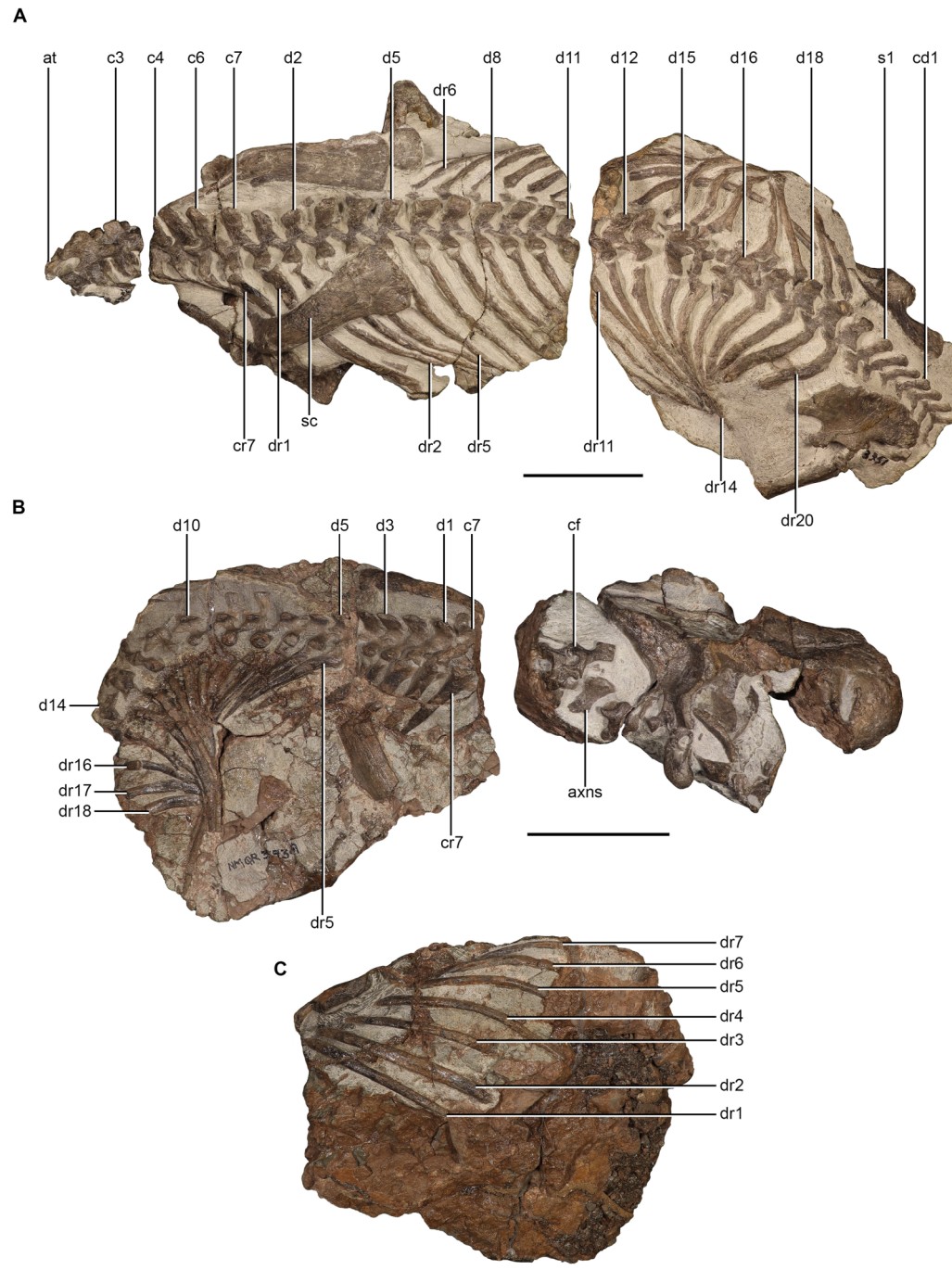

**Figure 1 Axial skeleton of *Moschorhinus kitchingi*.** NMQR 3351 in dorsal view (A), NMQR 3939 in dorsal (B) and ventral (C) views. Scale bars equals 100 mm. Abbreviations: at, atlas; axns, neural spine of the axis; c, cervical vertebrae; cd, caudal vertebrae; cf, cervical fragments; cr, cervical rib; d, dorsal vertebrae; dr, dorsal rib; s, sacral vertebrae; sc, scapula. Photographs by Brandon P. Stuart.

The axis bears a dorsoventrally tall, anteriorly extended, and robust neural spine that is hatchet shaped in lateral view (Figs. 2C, 2D, 2G and 2H). In dorsal view, the anterior end of the neural spine originates as a sharp point, which expands mediolaterally increasing width

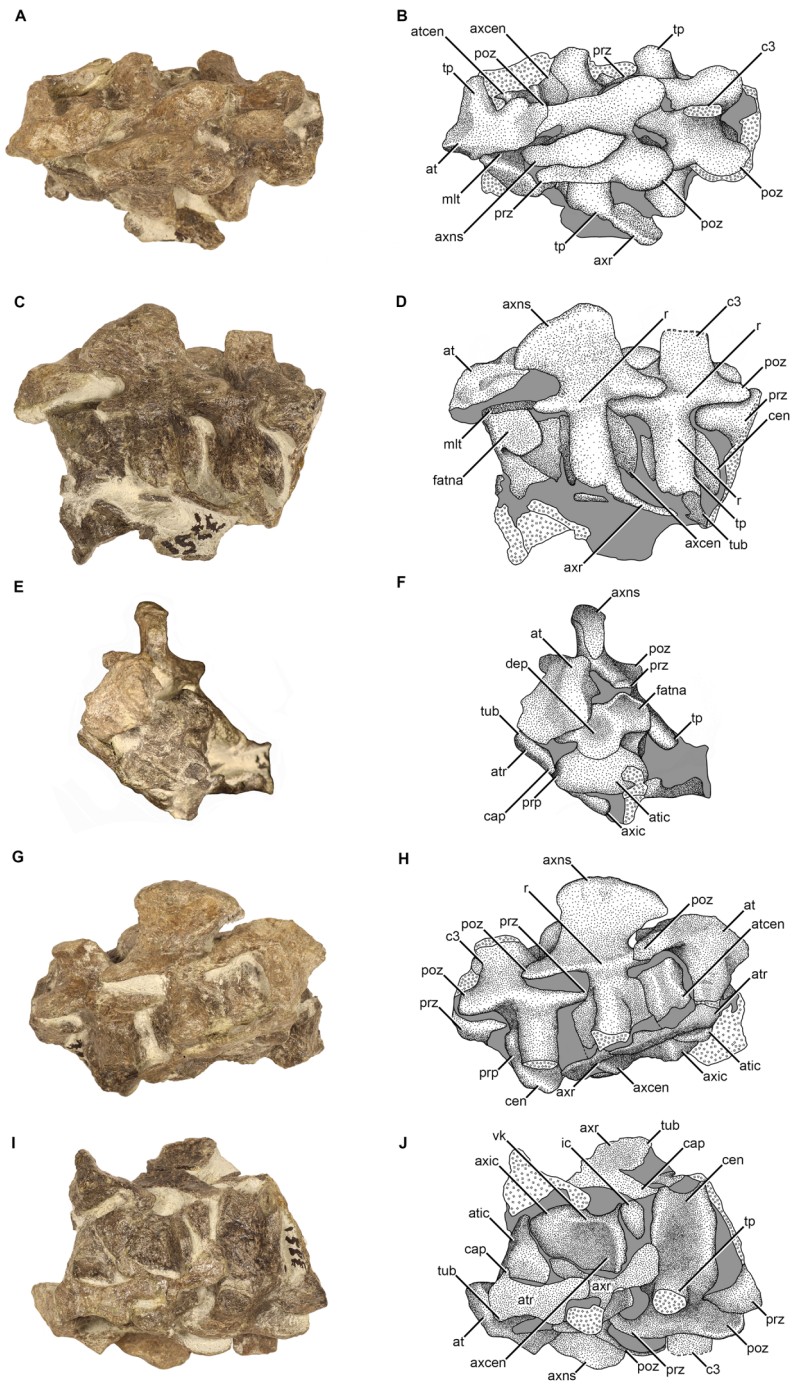

**Figure 2 Atlas-axis complex and third cervical of NMQR 3351.** Photographs and stipple drawing of the atlas-axis complex of NMQR 3351 in dorsal (A and B), left lateral (C and D), anterior (E and F), right lateral (G and H), and ventral views (I and J). Grey colouration indicates matrix and hatching indicates damaged surfaces. Scale bar equals 50 mm. Abbreviations: at, atlas; atcen, atlantal centrum; atic, atlantal intercentrum; atr, atlantal rib; axcen, axial centrum; axic, axial intercentrum; axns, axial neural spine; axr, axial rib; c, cervical vertebrae; cap, capitulum; cen, centrum; dep, depression; fatna, facet for atlas neural arch; ic, intercentrum; mlt, midline trough; poz, postzygapophysis; prp, parapophysis; prz, prezygapophyses; r, ridge; tp, transverse process; tub, tuberculum; vk, ventral keel. Photographs and illustrations by Brandon P. Stuart.

posteriorly (Figs. 2A and 2B). The greatest mediolateral expansion occurs approximately two-thirds down the length of the spine, from this point the spine narrows abruptly to terminate as a sharp point. The anterior extension is best seen in lateral view where it overhangs the anterior end of the neural arch (Figs. 2C, 2D, 2G and 2H) similar to that described for *Ericiolacerta* (*Watson, 1931*), but differs from the "backwardly directed" neural spine described for *Mirotenthes* (*Attridge, 1956*: 76) and the posteriorly expanded neural spine described for *Microgomphodon* (*Abdala et al., 2014a*). The posterior margin of the neural spine is relatively straight and slightly anterodorsally inclined. The lateral surface is anteroposteriorly long and dorsoventrally concave. The concavity of the lateral lamina slopes ventrolaterally, protruding laterally at the base of the neural arch where it forms a prominent anteroposterior lateral ridge, which runs longitudinally along the base of the neural arch into the lateral margins of the pre-and postzygapophyses (Figs. 2C, 2D, 2G and 2H). A complete neural spine, which is interpreted as the neural spine of the axis, is preserved in NMQR 3939 in close association with the cervical fragments preserved posterior to the skull (Fig. 1B). The neural spine is exposed in lateral view and is similar to the axial neural spine of NMQR 3351 by being anteriorly expanded at the dorsal end and dorsoventrally concave on the lateral surface (Fig. 1B).

The axial neural arch is approximately trapezoidal in shape in dorsal view, with its anterior end being markedly mediolaterally narrower than the broader posterior end (Figs. 2A and 2B). The neural arch is fused to the centrum with no neurocentral fusion lines being visible. The prezygapophyses are mediolaterally narrow, peg-like, horizontally orientated, and are separated medially by a deeply concave sulcus (Figs. 2A and 2B). The postzygapophyses are mediolaterally broader, horizontally orientated, and separated medially by a comparatively broader and deeply concave sulcus (Figs. 2A and 2B). The postzygapophyses are considerably larger than the prezygapophyses, similar to those described for *Olivierosuchus* (*Botha-Brink & Modesto, 2011*), but differs from *Ericiolacerta* (*Watson, 1931*) in which the prezygapophyses are larger than the postzygapophyses. No anapophyses are present. The transverse processes emerge below the base of the neural arch, anteriorly, closer to the prezygapophyses (Figs. 2C, 2D, 2G and 2H). The transverse processes are robust and project ventrolaterally similar to those described for *Olivierosuchus* (*Fourie & Rubidge, 2007*; *Botha-Brink & Modesto, 2011*).

The axial centrum is robust and amphicoelous (Figs. 2C and 2D). The lateral surface is deeply anteroposteriorly concave and the posterior margin is boarded by a prominent flat dorsoventral rim (Figs. 2C and 2D). The axial intercentrum is a large wedge-shaped bone and is fused to the anterior border of the axial centrum (no suture line is visible) (Figs. 2G–2J). The fusion of the axial intercentrum to the axial centrum forms a conspicuous anterior projection on which the atlantal centrum lies upon. The lateral surfaces of the axial centrum slope medially to meet at the midline off the ventral surface to form a flat ventral keel that runs anteroposteriorly across the centrum (Figs. 2I and 2J). The posteroventral margin of the centrum is concave, which provides accommodation for the following intercentrum (Figs. 2I and 2J).

Both of the axial ribs are preserved (Fig. 2), but they are damaged. The right axial rib is deformed and obscured by the right atlantal rib (Figs. 2I and 2J). The proximal end of the

tuberculum of the left axial rib is still attached to the left transverse process of the axis (Figs. 2A and 2B). The rest of the tuberculum, capitulum, and shaft has broken off and rotated approximately 180° along the sagittal plane so that the distal end of the shaft is projecting anteriorly and the dorsal surface of the rib is facing ventrally (Figs. 2I and 2J). The axial rib is dichocephalous and mediolaterally thin. The tuberculum is dorsoventrally tall and the capitulum is comparatively low (Figs. 2I and 2J). The third intercentrum is preserved posterior to the axial centrum (Figs. 2I and 2J). It is essentially a small wedge-shaped bone and is markedly smaller than the atlantal and axial intercentra. The intercentra are similar to those described for *Olivierosuchus* (*Botha-Brink & Modesto, 2011*), *Tetracynodon darti* (*Sigurdsen et al., 2012*), and *Gorynychus masyutinae* (*Kammerer & Masyutin, 2018a*).

**Postaxial cervical series**—The third cervical is preserved still in articulation with the axis (Figs. 1A and 2) whereas the rest of the cervical series is preserved on the second block of NMQR 3351 (Fig. 1A). The third cervical is slightly displaced relative to the axis and the distal end of the neural spine has broken off. The fourth cervical is damaged due to the break. The fifth cervical is well preserved and the neural spine of the sixth cervical is deformed due to a crack. The seventh vertebra of NMQR 3351 is well preserved and is transitional between the cervical and dorsal series. The first vertebrae preserved on the second block of NMQR 3939 is interpreted as the seventh cervical, but it is badly damaged (Fig. 1B).

The neural spines of the fifth and sixth cervical are dorsoventrally tall, anterodorsally inclined, and markedly shorter anteroposteriorly than that of the axial neural spine (Figs. 3A and 3B). They are approximately rectangular in lateral view with straight anterior and posterior margins so that their anteroposterior length is consistent throughout the height of the spine. The neural spine of the seventh cervical is directed more dorsally and bears weakly convex and concave anterior and posterior margins respectively (Figs. 3A and 3B). The lateral ridge of the neural arch is still present on the third, fifth, and sixth cervical, but is less pronounced than the lateral ridge of the axis, and is absent on the seventh (Figs. 2C, 2D, 2G, 2H, 3A and 3B). The pre- and postzygapophyses of the third, fifth, and sixth cervical are mediolaterally broad and horizontally orientated (Figs. 2C, 2D, 2G, 2H, 3A and 3B). The prezygapophysis of the seventh cervical is similar to those of the preceding cervicals, but the postzygapophysis is orientated more vertically (Figs. 3A and 3B). The transverse processes of the third, fifth, and sixth cervical are mediolaterally short, anteroposteriorly narrow, ventrolaterally directed, and emerge more anteriorly than posteriorly from below the lateral ridge of the neural arch (Figs. 3A and 3B) similar to those described for *Scaloposaurus* (*Kemp, 1986*). In contrast, the transverse process of the seventh cervical emerges more dorsally, closer to the level of the neural arch, projects more laterally, and is more robust being anteroposteriorly broader (Figs. 3A and 3B). The lateral surface of the centrum of the third cervical is deeply anteroposteriorly concave. The lateral surfaces meet at the midline of the ventral surface to form a sharp ventral keel (Figs. 2C, 2D, 2G, 2H, 2I and 2J) similar to the cervical centra described for other therocephalians (*Botha-Brink & Modesto, 2011*; *Sigurdsen et al., 2012*; *Abdala et al., 2014b*, *2014a*), but

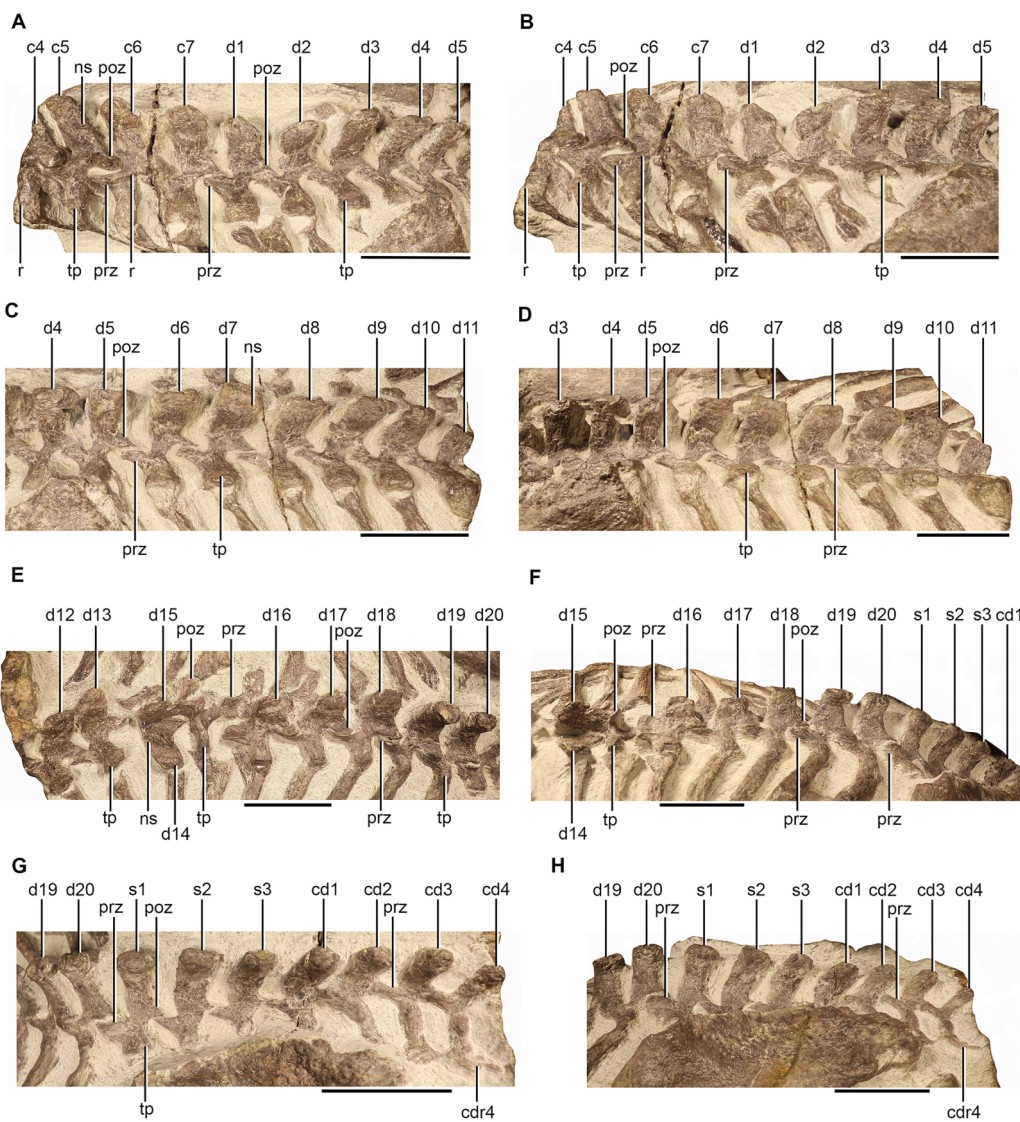

**Figure 3 Vertebrae of NMQR 3351.** Posterior cervical and anterior dorsal vertebrae in dorsolateral (A) and left lateral (B) views, anterior and mid-dorsal vertebrae in dorsolateral (C) and left lateral (D) views, mid- and posterior dorsal vertebrae in dorsolateral (E) and left lateral (F) views, and sacral and caudal vertebrae in dorsolateral (G) and left lateral (H) views. Scale bars all equal 50 mm. Abbreviations: c, cervical vertebrae; cd, caudal vertebrae; cdr, caudal rib; d, dorsal vertebrae; ns, neural spine; poz, post-zygapophysis; prz, prezygapophysis; r, ridge; s, sacral vertebrae; tp, transverse process. Photographs by Brandon P. Stuart.                                                 

differing from the flat ventral keel of the axis. The anterior margin slants posteriorly approximately halfway of the height of the centrum, shortening the anteroposterior length at the ventral end. The lateral surfaces of the posterior margin bear a flat parapophysis (Figs. 2G and 2H).

### Dorsal vertebrae

The dorsal vertebrae of NMQR 3351 are well preserved (Figs. 3A–3I), apart from the 11th dorsal which has suffered damage due to the transverse break between the second and

third block (Figs. 3C and 3D). The mid-posterior dorsals (d13–d15) are disarticulated so that the 14th dorsal is rotated approximately 90° laterally to the left and the 15th dorsal is rotated approximately 90° anteriorly (Figs. 3E and 3F). The anterior dorsals (d1–d4) of NMQR 3939 are relatively well preserved, but the neural spines of the mid-dorsals (d5–d10) are weathered and their transverse processes are deformed and the posterior dorsals (d11–d14) are too damaged to warrant any meaningful description.

The neural spines of the first and second dorsal of NMQR 3351 are dorsally directed and dorsoventrally low (Figs. 3A and 3B). The dorsal margin of the first dorsal slants posteroventrally, in contrast, the dorsal margin of the second dorsal inclines posterodorsally (Figs. 3A and 3B). The neural spines of the succeeding anterior and mid-dorsal vertebrae (d3–d15) are all dorsoventrally tall, mediolaterally narrow, and become increasingly posterodorsally curved moving posteriorly (Figs. 3A–3F) similar to that described for *Scaloposaurus* (*Kemp, 1986*). The neural spines of the 16th and 17th dorsal are dorsoventrally low and the neural spines of the final three dorsals (d18–d20) are dorsoventrally tall as those of the main dorsal region (d3–d15), but are mediolaterally broader, anteroposteriorly expanded at their base, and project dorsally (Figs. 3H and 3I).

The pre- and postzygapophyses of the anterior and mid-dorsals are vertically orientated and mediolaterally thin on their dorsal margins, particularly those of the anterior-mid-dorsals (d6–d13), giving them a more gracile appearance (Figs. 3A–3D). The pre- and postzygapophyses of the posterior dorsals (d15–d20) are more horizontally orientated and laterally expanded (Figs. 3E and 3F). The prezygapophyses of the anterior and posterior dorsals extend anteriorly and the postzygapophyses of the anterior and mid-dorsals (d1–d15) extend posteriorly (Figs. 3A–3F), but the postzygapophyses of the posterior dorsals (d16–d20) extend progressively more posterolaterally (Figs. 3E and 3F). The ventral margin of the neural spine of the disarticulated 14th dorsal diverges ventrolaterally halfway down the length of the spine to form the posterior margins of the postzygapophyses, which are separated by a deep and mediolaterally narrow fossa (Fig. 3E). The ventral surface of the postzygapophyses slant ventromedially (Figs. 3E and 3F). The medial articulatory surface of the prezygapophyses of the 16th dorsal extends ventrally from the mediolaterally sharp dorsal margin and slopes medially into a mediolaterally broad dorsal articulatory surface (Fig. 3E). The prezygapophyses are separated medially by a deep sulcus that terminates as a sharp point at the anterior margin of the neural arch (Fig. 3E).

The transverse processes of the anterior dorsals (d1–d5) project laterally from the level of the neural arch and project progressively more posterolaterally in the mid-dorsals (d6–d13), but less so in the posterior dorsals (d16–d20) (Figs. 3A–3F). The transverse process of the anterior and mid-dorsals (d1–d13) are anteroposteriorly broad and the posterolateral edge of the distal end becomes increasingly upturned in the mid-dorsals (d6–d13) (Figs. 3A–3F). The transverse processes of the posterior dorsals (d16–d20) are anteroposteriorly narrow and become mediolaterally reduced sequentially (Figs. 3E and 3F).

### Sacral vertebrae

The ribs of the sacral vertebrae of NMQR 3351 are not exposed and only the proximal portions of the transverse processes are exposed (Figs. 3G and 3H). The succeeding three vertebrae after the dorsals are identified as the sacrals based on the presence of three depressions on the medial surface of the ilium of SAM-PK-K10981 (see below).

The neural spines of the sacrals are dorsoventrally tall and rectangular in lateral view (Fig. 3H). The neural spine of the first sacral projects slightly posterodorsally and the second and third are posterodorsally curved (Fig. 3H). The prezygapophysis of the first sacral is robust and similar to those of the preceding posterior dorsals (Figs. 3G and 3H). The prezygapophyses of the second and third sacrals are slightly mediolaterally narrower (Fig. 3G). The proximal portion of the transverse process of the first sacral is anteroposteriorly broader than the transverse processes of the second and third sacral and bears a convex dorsal surface (Fig. 3H).

### Caudal vertebrae

The neural spines of the caudal vertebrae are posterodorsally curved and dorsoventrally lower and anteroposteriorly shorter than those of the sacrals and they reduce in height and length sequentially (Fig. 3H). The prezygapophyses are anteroposteriorly long whereas their postzygapophyses are anteroposteriorly short (Figs. 3G and 3H). The transverse processes are anteroposteriorly narrow, mediolaterally long, and extend posterolaterally from the base of the neural arch (Fig. 3G).

## Ribs

### Cervical ribs

Only the proximal ends of the right cervical ribs of NMQR 3351 are exposed (Fig. 4A). The cervical ribs are dichocephalous. The tubercula of the fourth, fifth, and sixth cervical rib are dorsoventrally tall and anteroposteriorly broad (Fig. 4A). The tuberculum of the seventh cervical rib is more robust than those of the preceding ribs by being dorsoventrally taller and anteroposteriorly broader (Fig. 4A). The capitulum of the fifth cervical rib is dorsoventrally low and slender (Fig. 4A). The proximal portion of the shafts are mediolaterally thin, posteroventrally directed, and become progressively anteroposteriorly broader from the fifth cervical rib (Fig. 4A). The distal ends of shafts of the fourth, fifth, and sixth cervical ribs are not exposed, but they are likely not very long as the cervical ribs are short in all therocephalian taxa in which they are known (*Fourie & Rubidge, 2007*, *2009*; *Botha-Brink & Modesto, 2011*; *Sigurdsen et al., 2012*). The shaft of the seventh cervical rib is long, indicated by the exposure of the distal end, which is lying under the distal end of the first dorsal rib (Fig. 1A).

### Dorsal ribs

All of the dorsal ribs of NMQR 3351 are preserved, but the right ribs of the anterior dorsals (d1–d5) are not exposed (Fig. 1A). NMQR 3939 preserves at least 17 right dorsal ribs on the dorsal side of the second block and seven left dorsal ribs on the ventral side (Figs. 1B, 1C, 4B and 4C).

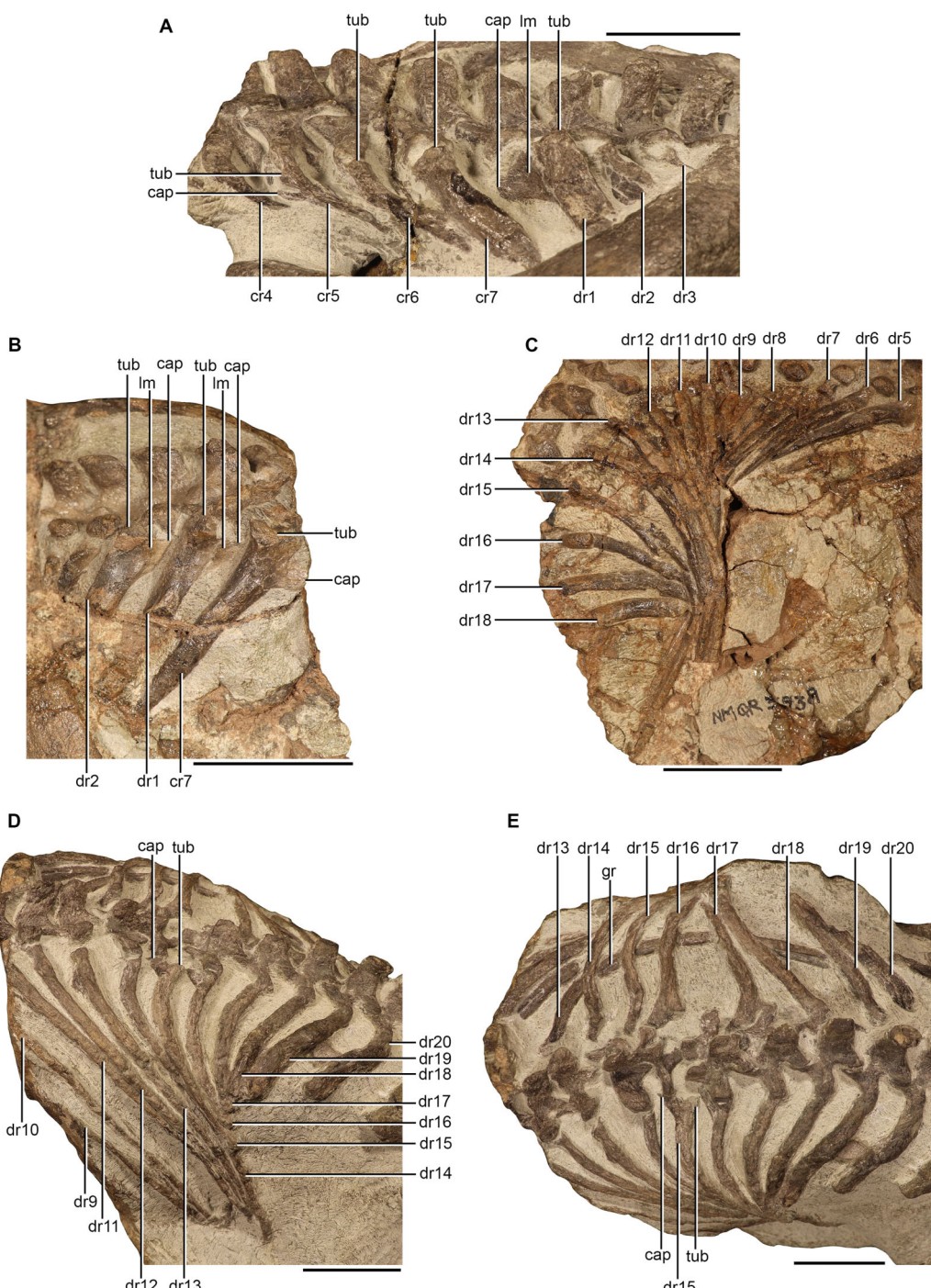

**Figure 4 Ribs of *Moschorhinus kitchingi*.** Cervical ribs of NMQR 3351 in left lateral view (A), cervical and anterior dorsal ribs of NMQR 3939 in right lateral view (B), dorsal ribs of NMQR 3939 in dorsal view (C), posterior dorsal ribs of NMQR 3351 in left lateral view (D), and posterior dorsal ribs of NMQR 3351 in dorsal view (E). Scale bars all equal 50 mm. Abbreviations: cap, capitulum; cr, cervical rib; dr, dorsal rib; gr, groove; lm, lamina; tub, tuberculum. Photographs by Brandon P. Stuart.

The proximal head of the first dorsal rib of NMQR 3351 bears an anteroposteriorly broad tuberculum similar to that of the seventh cervical rib (Fig. 4A). The proximal head is dichocephalous, but the tuberculum and capitulum are conjoined by a thin lamina, which forms a dorsoventrally tall and confluent synapophysis that is triangular in anterior view (Fig. 4A). A similar head condition is present in the first three dorsal ribs of NMQR 3939 (Fig. 4B). The tubercula of the proximal heads of the posterior dorsal ribs (d15–d17) of NMQR 3351 are mediolaterally reduced, forming a long pedicle with the ventrally expanded capitulum lying below, creating a strong holocephalous appearance (Figs. 4D and 4E). The shafts of the anterior and mid-dorsal (d1–d13) ribs of NMQR 3351 are long and bear a sharp dorsal margin (Fig. 1A). The posterior surface of the mid dorsal ribs (d8–d13) is flat and a thin longitudinal costal groove, which extends to approximately two-thirds of the ribs' length, is present (Fig. 1A) similar to that described for *Olivierosuchus* (*Fourie & Rubidge, 2007*) and *Promoschorhynchus* (*Huttenlocker, Sidor & Smith, 2011*). The shafts of the mid-posterior dorsal ribs (d15–d20) shorten sequentially, with the posterior dorsal ribs (d18–d20) being markedly shorter (Figs. 4D and 4E). The last three posterior dorsal ribs are fused to their respective transverse processes and their shafts are dorsoventrally flattened and anteroposteriorly broader than preceding dorsal ribs (Figs. 4D and 4E).

### Caudal ribs

The proximal heads of the two right ribs of the last two caudal vertebrae of NMQR 3351 are preserved, but they are not well exposed (Figs. 3G and 3H). They are fused to their respective transverse processes and have anteroposteriorly short shafts that are posteriorly projected similar to those described for early-diverging therocephalians (*Cys, 1967*; *Fourie & Rubidge, 2009*) and *Olivierosuchus* (*Fourie & Rubidge, 2007*).

## Scapula

Scapulae are preserved to varying degrees in NMQR 48, NMQR 3939, BP/1/4227, and NMQR 3351. However, they are best preserved in NMQR 48, BP/1/4227, and NMQR 3351 (Figs. 5A–5L). NMQR 48 preserves the complete ventral end and a small section of the midshaft of the left scapula of which the anterior and lateral views have been exposed (Figs. 5A and 5B). BP/1/4227 preserves the complete left scapula as a separate element allowing for the description of all views (Figs. 5C–5F). NMQR 3351 preserves both the complete left and right scapulae (Figs. 5G–5L). The entirety of the lateral surface of both scapular blades is exposed. The anterior and posterior sides of the ventral ends are slightly exposed, with the right being more exposed than the left. Measurements of the scapulae are given in Table 2.

The scapula is a dorsoventrally elongated, curved bone comprised of a slender blade, a semi-circular shaft-like middle region, and a robust ventral end. In NMQR 3351, BP/1/4227, and NMQR 3939 the dorsal end is mediolaterally constricted and anteroposteriorly expanded as in other therocephalians (*Huttenlocker, Sidor & Smith, 2011*). The dorsal margin of the blade is only slightly anteroposteriorly convex, and the anterior and posterior margins are slightly rounded. The most dorsal end of the left scapula of NMQR

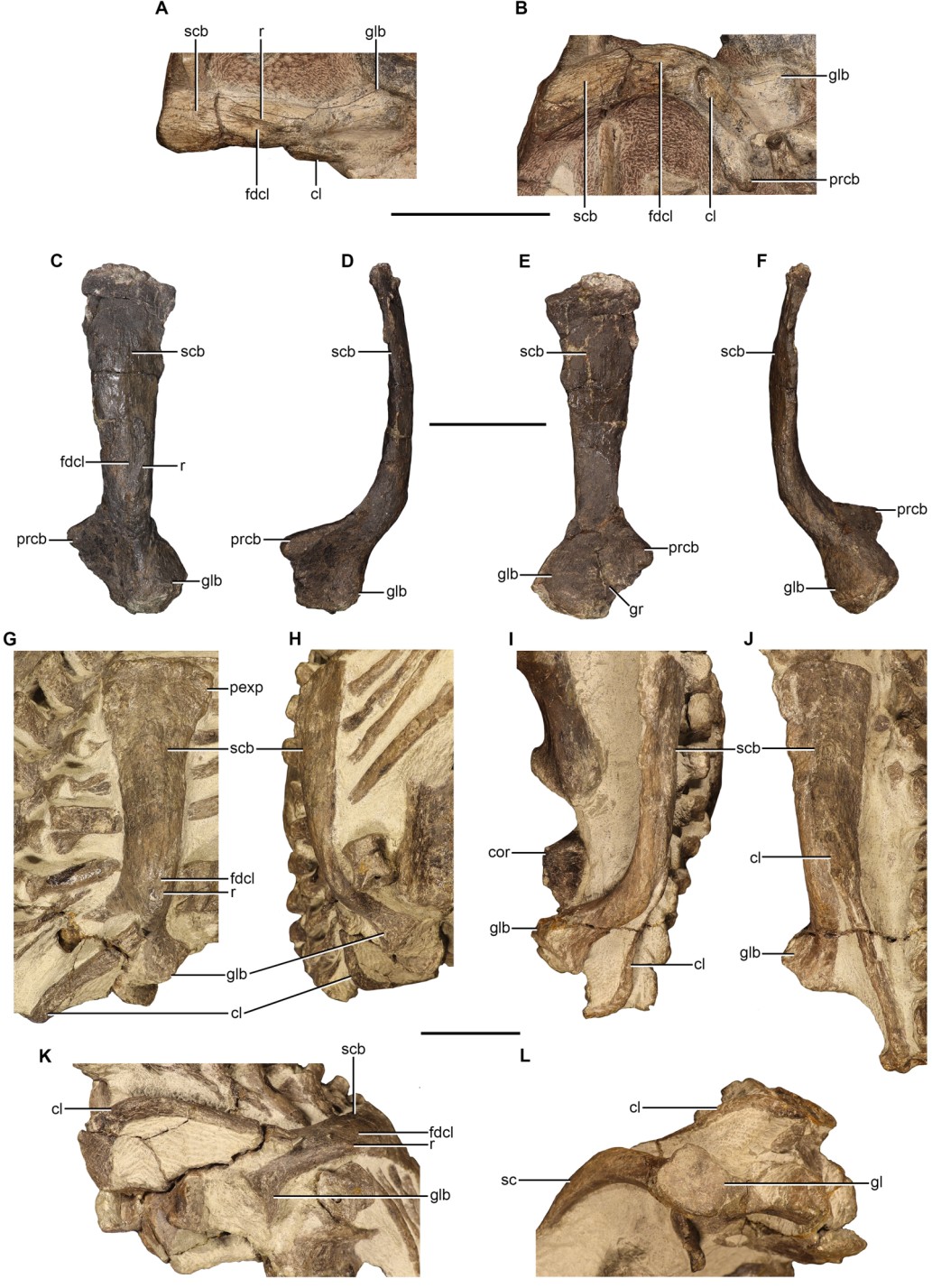

**Figure 5 Scapulae of *Moschorhinus kitchingi*.** Left scapula of NMQR 48 in lateral (A) and anterior (B) view, left scapula of BP/1/4227 in lateral (C), anterior (D), medial (E), and posterior (F) views, left scapula of NMQR 3351 in lateral (G) and posterior (H) views, right scapula of NMQR 3351 in posterior (I) and lateral (J) views, the left (K) and right (L) glenoid of NMQR 3351. Scale bars all equal 50 mm. Abbreviations: cl, clavicle; cor, coracoid; fdcl, fossa for the dorsal end of the clavicle; gl, glenoid; glb, glenoid buttress; gr, groove; pexp, posterior expansion of the scapula; prcb, procoracoid buttress; r, ridge; sc, scapula; scb, scapular blade. Photographs by Brandon P. Stuart.

**Table 2 Measurements of the pectoral girdle elements of specimens of *Moschorhinus kitchingi*.**

| | NMQR 48 | | NMQR 3939 | BP/1/4227 | CGS GHG299 | NMQR 3351 | |
| --- | --- | --- | --- | --- | --- | --- | --- |
| | Left | Right | Left | Left | | Left | Right |
| **Scapula** | | | | | | | |
| Height | | 86.48* | 87.51* | 148.66 | – | 168.42* | 170.75* |
| Length of dorsal end | | – | 28.07* | 40.12* | – | 60.76 | 53.17* |
| Length of ventral end | | 45.96* | – | 54.69 | – | 57.46* | – |
| Width of glenoid | | – | – | 31.60* | – | – | 37.12 |
| **Procoracoid** | | | | | | | |
| Length | – | | 47.41* | – | – | – | 62.45* |
| **Coracoid** | | | | | | | |
| Length of blade | 39.98* | 34.87* | – | – | – | – | 69.19* |
| Length of glenoid | 16.78* | 21.37* | – | – | – | – | – |
| **Interclavicle** | | | | | | | |
| Length | 38.78* | – | – | – | – | 70.03 | |
| Width of anterior end | – | – | – | – | – | – | |
| Width of posterior end | 25.63 | – | – | – | – | 33.10 | |
| **Sternum** | | | | | | | |
| Length | 62.24* | – | – | – | 86.31* | 93.55* | |
| Width | 62.12 | – | – | – | 84.55 | 92.68 | |
| Length of vmr | 21.71 | – | – | – | 32.99 | 44.96 | |

Note:
* Indicates element is partially covered by matrix or incomplete.

3351 possesses a conspicuous posterior expansion (Fig. 5G). This is not present on any other scapula; however, all of the other preserved scapulae are damaged along their posterodorsal margin. The lateral surface of the most dorsal end of the blade is slightly anteroposteriorly convex and the bone surface of NMQR 3351 and BP/1/4227 is extremely rough exhibiting prominent longitudinal muscle scars (Figs. 5C and 5G). The medial surface of the scapula is only visible in BP/1/4227 and it is relatively flat (Fig. 5E).

The scapular blade constricts anteroposteriorly approximately one-third from the dorsal margin becoming semi-circular in cross-section by developing a convex lateral surface and remaining flat on the medial surface. The convexity of the lateral surface continues ventrally and forms a broad dorsoventral ridge that extends to the ventral margin of the scapula. The broad ridge divides the distal half of the scapula into an anterior and posterior surface, with the former having a steeper slope than the latter. This is similar to other eutherocephalians such as *Mirotenthes* (*Attridge, 1956*), *Olivierosuchus* (*Fourie & Rubidge, 2007*; *Botha-Brink & Modesto, 2011*), and *Ictidosuchoides* (*Fourie, 2013*), but differs from the sharp dorsoventral ridge that has been described for *Microgomphodon* (*King, 1996*). The scapula projects medially with a sharp curve approximately two-thirds from the dorsal margin (Figs. 5D, 5F, 5H and 5I). A rough fossa that is bounded by a sharp ridge on its posterior border is present on the lateral surface of

NMQR 48, BP/1/4227, and NMQR 3351 for the attachment of the clavicle (Figs. 5A–5C, 5G and 5K).

The ventral end of the scapula consists of a mediolaterally thin, plate-like, triangular procoracoid buttress and a robust, transversely expanded glenoid buttress (Figs. 5A–5L). The procoracoid buttress and glenoid buttress are separated on the lateral surface by the broad ridge described above (Fig. 5C). The anterior surface of the procoracoid buttress of BP/1/4227 is smooth and weakly depressed (Figs. 5C and 5D). The posterior surface of the glenoid buttress of BP/1/4227 is weakly convex (Figs. 5C and 5F), but weakly concave and raised on the posteromedial margin in the right scapula of NMQR 3351 (Figs. 5H and 5I). No distinct protuberance on the posterior surface of the glenoid buttress is observed on any of the studied scapulae as described for 'Zinnosaurus' paucidens and Simorhinella (Boonstra, 1964; Abdala et al., 2014b). On the medial surface of BP/1/4227, the procoracoid buttress and glenoid buttress are separated by a deep groove (Fig. 5E) differing from that described for Simorhinella in which a ridge is present in this area (Abdala et al., 2014b). The glenoid surface of the right scapula of NMQR 3351 is approximately square in outline in ventral view and is weakly concave (Fig. 5K). In contrast, the glenoid surface is convex in BP/1/4227, but this surface has suffered damage.

An ossified cleithrum is not present in any specimen studied, but has been reported for Ericiolacerta (Watson, 1931), Pristerognathus (Broom, 1936), an undescribed akidnognathid specimen (Huttenlocker, Sidor & Smith, 2011: 415), and Ictidosuchoides (Fourie, 2013). None of the studied scapulae show any indication for the attachment of a cleithrum on the anterior margin of the blade. A facet for the attachment for a cleithrum on the anterior margin of the scapular blade has been reported for 'Zinnosaurus' paucidens (Boonstra, 1964). No acromion process is present in any of the scapulae as described for other therocephalian genera (Kemp, 1986; Fourie & Rubidge, 2007; Botha-Brink & Modesto, 2011).

## Coracoid

Coracoids are only preserved in NMQR 48 and NMQR 3351. Both of the coracoids in NMQR 48 are preserved and they are exposed in lateral view (Figs. 6A and 6B). The majority of the posterior and ventral margins of the right coracoid are covered by matrix as are the overlying clavicle and interclavicle. The exposed surface of the right coracoid suggests it has been deformed when compared to the left coracoid which is better preserved. Only the right coracoid is preserved in NMQR 3351, but it is broken and missing the dorsal end (Fig. 6C).

The coracoid is a blade-like bone with a distinct, robust dorsal neck. In NMQR 48, the dorsal neck of the coracoids are posteriorly expanded, transversely thickened, and exhibit a rugose convex surface, which forms the ventral surface of the glenoid (Figs. 6A and 6B). The blades of the coracoids in NMQR 48 and NMQR 3351 are mediolaterally thin and flat (Figs. 6A–6C). In NMQR 48 the blades extend ventrally as well as posteriorly and attenuate to a distinct rounded tuberosity as described for Jiufengia (Liu & Abdala, 2019). The ventral margins of the blades are slightly convex, and the anterior margins are obscured, but presumably they would have been relatively straight to form the procoracoid-coracoid

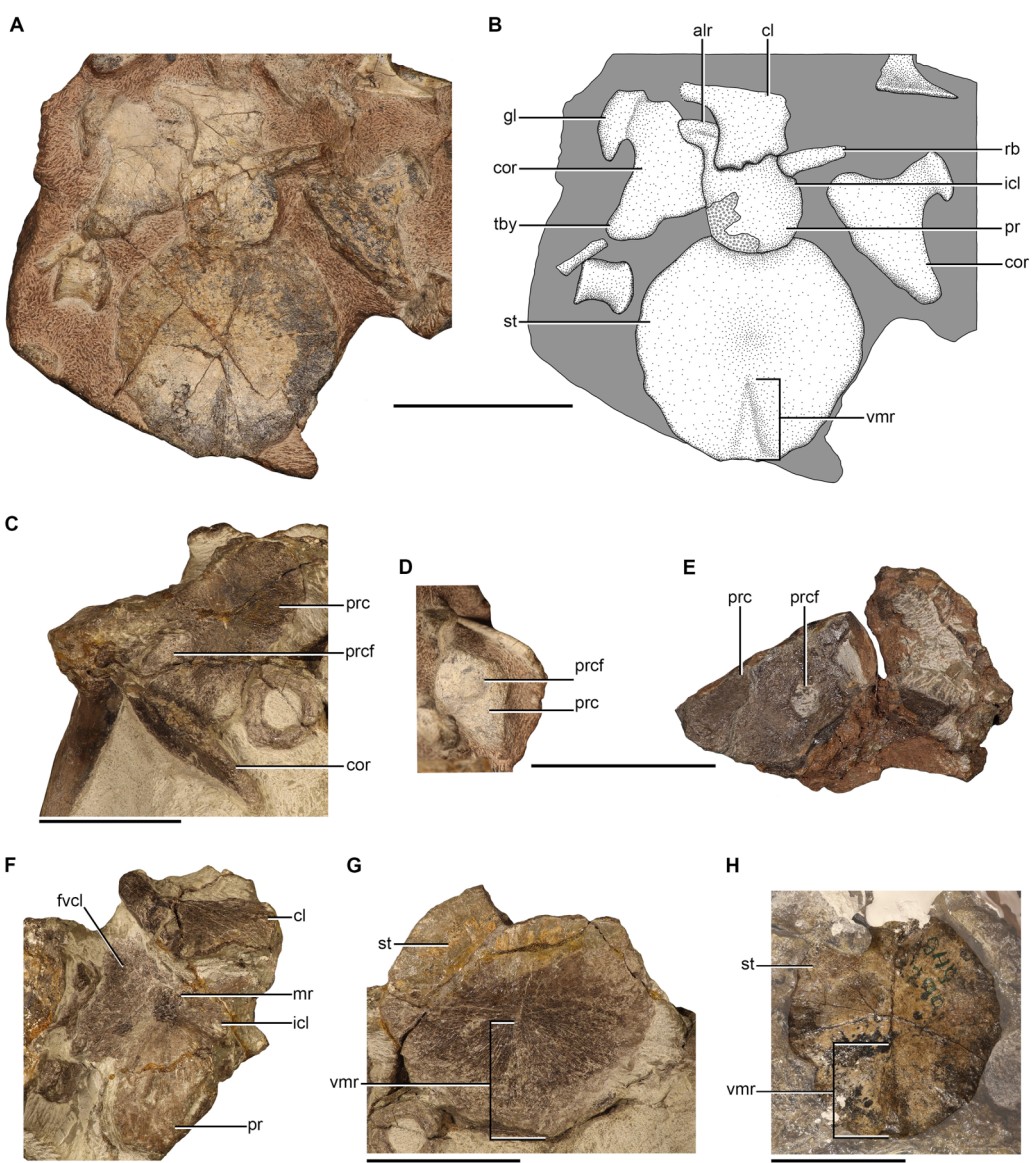

**Figure 6 Pectoral girdle elements of *Moschorhinus kitchingi*.** Photograph (A) and stipple drawing (B) of the pectoral elements of NMQR 48 in ventral view. Grey colouration indicates matrix and hatching indicates damaged surfaces. Right coracoid and procoracoid (medial view) of NMQR 3351 (C), right procoracoid of NMQR 48 in medial view (D), right procoracoid of NMQR 3939 in medial view (E), interclavicle of NMQR 3351 in ventral view (F), sternum of NMQR 3351 in ventral view (G), and sternum of CGS GHG299 (semi-transparent background) in ventral view (H). Scale bars equal 50 mm. Abbreviations: alr, anterolateral ramus of the interclavicle; cl, clavicle; cor, coracoid; fvcl, fossa for the ventral end of the clavicle; icl, interclavicle; gl, glenoid; mr, median ridge of the interclavicle; pr, posterior ramus of the interclavicle; prc, procoracoid; prcf, procoracoid foramen; rb, rib ;st, sternum; tby, tuber-osity; vmr, ventromedial ridge of the sternum. Photographs and illustration by Brandon P. Stuart.

suture. *Watson (1931)*, *Attridge (1956)*, and *Boonstra (1964)* noted that the coracoids of *Ericiolacerta*, *Mirotenthes*, and early-diverging therocephalians were comparatively smaller than the procoracoids. The only specimen that preserves the two respective elements is NMQR 3351, although they are disassociated and slightly damaged, they

appear to be of similar size (Fig. 6C). Apart from this, the coracoids of NMQR 48 and NMQR 3351 are similar to those described for other therocephalian genera (*e.g.*, *Kemp, 1986*; *Botha-Brink & Modesto, 2011*; *Liu & Abdala, 2019*).

## Procoracoid

Procoracoids are preserved in NMQR 48, NMQR 3939, and NMQR 3351. Only the medial surface of the most posterior portion of the right procoracoid is exposed in NMQR 48 (Fig. 6D). NMQR 3939 preserves the right procoracoid and is exposed in medial view, but it is damaged along its posterodorsal and posteroventral margins (Fig. 6E). NMQR 3351 only preserves the right procoracoid, which is still closely articulated with the right scapula (Fig. 6C).

The procoracoid is a quadrangular plate-like bone with a flat medial surface. The anterior margin is convex whereas the posterodorsal and posteroventral margins, which would have contacted the procoracoid buttress of the scapula and the coracoid respectively, are straight as described for other therocephalians (*Attridge, 1956*; *Cys, 1967*; *Kemp, 1986*; *Botha-Brink & Modesto, 2011*; *Huttenlocker, Sidor & Smith, 2011*; *Sigurdsen et al., 2012*). The posterodorsal and posteroventral margins converge to meet at the glenoid in the form of a sharp point. Although no specimen preserves an articulated scapulocoracoid, it appears that the procoracoid does not contribute to the articulatory surface of the glenoid as described for other therocephalians (*Boonstra, 1964*; *Botha-Brink & Modesto, 2011*; *Sigurdsen et al., 2012*) with the exception of the scylacosaurid described by *Cys (1967)* in which the procoracoid contributes to the anterodorsal surface of the glenoid.

A large, circular procoracoid foramen that is entirely enclosed by the procoracoid is present in all the studied procoracoids (Figs. 6C–6E), and it is positioned close to the posterolateral margin as described for early-diverging therocephalians (*Boonstra, 1964*; *Cys, 1967*; *Fourie & Rubidge, 2009*), as well as for the eutherocephalians *Ictidosuchus* (*Broom, 1900*), *Mirotenthes* (*Attridge, 1956*), *Olivierosuchus* (*Botha-Brink & Modesto, 2011*), *Promoschorhynchus* (*Huttenlocker, Sidor & Smith, 2011*), *Tetracynodon darti* (*Sigurdsen et al., 2012*), *Ictidosuchoides* (*Fourie, 2013*) and *Regisaurus* (BP/1/5394; B. Stuart, 2022, personal observation). This differs from the more derived eutherocephalians *Ericiolacerta* (*Watson, 1931*) and *Scaloposaurus* (*Kemp, 1986*) in which the foramen is present between the procoracoid-coracoid suture as well as *Bauria* in which it is positioned on the scapula-procoracoid suture (*Watson, 1931*).

## Clavicle

Clavicles are only preserved in NMQR 48 and NMQR 3351. Both clavicles are preserved in NMQR 48, but they are incomplete. The left clavicle is overlying the right scapula, but is incomplete and only represented by a small portion of the dorsal end (Figs. 5A and 5B). The right clavicle is incomplete, missing the dorsal end and the majority of the shaft. The ventral end is well preserved and is still slightly articulated with the interclavicle (Figs. 6A and 6B). Both the clavicles are preserved in NMQR 3351. The left clavicle is incomplete, missing its dorsal end. The majority of the shaft and the ventral end is preserved, but it is

broken along its medial margin (Figs. 5G, 5H, 5K and 6F). The right clavicle is incomplete, missing its ventral end. The majority of the shaft and the dorsal end, which is still articulated with the right scapula, is preserved (Figs. 5I, 5J and 5L).

The clavicle is an elongated, fairly slender, curved element that is expanded at its dorsal and ventral ends which are connected by a narrow rod-like shaft. The dorsal ends of the clavicles of NMQR 48 and NMQR 3351 are flat and spatulate and are only slightly more transversely expanded than the shaft (Figs. 5B and 5J). The shafts of the clavicles of NMQR 3351 curve ventromedially and posteriorly at their ventral ends (Figs. 5H, 5I, 5K and 5L). The ventral ends of the clavicles of NMQR 48 and NMQR 3351 flare out into a dorsoventrally thin fan-like blade with a prominent posterior projection (Figs. 5K, 6A, 6B and 6F). The medial and posterior margins of the ventral end of the right clavicle in NMQR 48 are well preserved and exhibit prominent scalloped edges (Figs. 6A and 6B).

### Interclavicle

Interclavicles are preserved in NMQR 48 and NMQR 3351. The ventral surface of the interclavicle of NMQR 3351 is exposed, but has suffered damage to its outer margins and the left anterior end is obscured by the ventral end of the left clavicle (Fig. 6F). The interclavicle of NMQR 48 is better preserved, but likewise the majority of the anterior end is obscured by the ventral end of the right clavicle, and the left anterolateral surface is damaged by the distal end of a rib (Figs. 6A and 6B).

The interclavicle is a small, dorsoventrally thin, flat, plate-like bone. It is anteroposteriorly short and mediolaterally broad at its anterior end in contrast to the interclavicles described for early-diverging therocephalians, which are anteroposteriorly long, mediolaterally narrow, and approximately spoon-shaped in outline (Boonstra, 1964; Fourie & Rubidge, 2009). The complete outline of the interclavicle is obscured in both NMQR 3351 and NMQR 48, but it is approximately shield-shaped, with small laterally projecting rami on either side of the anterior end and a posterior projecting ramus (Figs. 6A, 6B and 6F), similar to the interclavicle described for Scaloposaurus (Kemp, 1986) and Promoschorhynchus (Huttenlocker, Sidor & Smith, 2011). This differs from the cruciform interclavicle described for Ericiolacerta (Watson, 1931) and Olivierosuchus (Fourie & Rubidge, 2007; Botha-Brink & Modesto, 2011), which appear to have mediolaterally wide projecting lateral rami.

Although much of the anterior end of NMQR 3351 is obscured, a prominent median ridge that extends ventrally to centre of the interclavicle, is observed (Fig. 6F) similar to that described for Ericiolacerta (Watson, 1931), Scaloposaurus (Kemp, 1986) and Olivierosuchus (Botha-Brink & Modesto, 2011). The right anterolateral surface of the interclavicle of NMQR 3351 is depressed, forming a fossa for the attachment of the ventral end of the clavicle (Fig. 6F) as described for Olivierosuchus (Botha-Brink & Modesto, 2011). The posterior ramus of NMQR 3351 and NMQR 48 is mediolaterally broad and the width is consistent throughout its length (Figs. 6A, 6B and 6F). The posterior ramus of NMQR 3351 and NMQR 48 terminates as a blunt point in contrast to Olivierosuchus (Fourie & Rubidge, 2007; Botha-Brink & Modesto, 2011), which terminates as a sharp point. The ventral surface of the posterior ramus of NMQR 48 is damaged, which has led to an

unnatural raised lineation (Figs. 6A and 6B). The ventral surface of the posterior end of NMQR 3351 is well preserved and is smooth as described for *Ericiolacerta* (*Watson, 1931*) and *Scaloposaurus* (*Kemp, 1986*), but contrasts with *Promoschorhynchus* (*Huttenlocker, Sidor & Smith, 2011*), which exhibits a well-developed midline ridge on the posterior ramus.

## Sternum

Sterna are preserved in NMQR 48, NMQR 3351, and CGS GHG299. The sternum of NMQR 48 is preserved in its entirety and does not appear to exhibit any deformation (Figs. 6A and 6B). The sternum in NMQR 3351 is mostly complete, however it has suffered damage from breakages and as a result a fragment of the left anterolateral end is missing, although the damage does not appear to have deformed it from its natural shape (Fig. 6G). The sternum of CGS GHG299 is complete and well preserved, but a portion of the anterior end is covered with protective plaster (Fig. 6H).

The sternum is a well-ossified, large plate-like bone as described for other eutherocephalians such as *Ericiolacerta* (*Watson, 1931*), *Olivierosuchus* (*Fourie & Rubidge, 2007*; *Botha-Brink & Modesto, 2011*), *Promoschorhynchus* (*Huttenlocker, Sidor & Smith, 2011*), *Tetracynodon darti* (*Sigurdsen et al., 2012*), and *Ictidosuchoides* (*Fourie, 2013*). This contrasts with early-diverging therocephalians where no specimen has been reported as preserving an ossified sternum and it is presumed to have been cartilaginous (*Broom, 1938*; *Boonstra, 1964*; *Fourie & Rubidge, 2009*). The sterna of NMQR 48, NMQR 3351, and CGS GHG299 are circular in outline (Figs. 6A, 6B, 6G and 6H) with their length being approximately equal to their width (Table 2). Variable sternal shapes have been reported for eutherocephalian taxa such as oval for *Olivierosuchus* (*Fourie & Rubidge, 2007*; *Botha-Brink & Modesto, 2011*) and *Promoschorhynchus* (*Huttenlocker, Sidor & Smith, 2011*), polygonal for *Tetracynodon darti* (*Sigurdsen et al., 2012*), and diamond-shaped for *Ictidosuchoides* (*Fourie, 2013*).

The anterior end of NMQR 48 is weakly depressed for the reception of the interclavicle (Figs. 6A and 6B) and it appears that the contact between these bones is weak, similar to *Promoschorhynchus* (*Huttenlocker, Sidor & Smith, 2011*), but contrasting to the condition in two specimens of *Olivierosuchus* (BP/1/3973 and BP/1/3849) in which the interclavicle and sternum are fused. The lateral sides of the ventral surface of NMQR 48, NMQR 3351, and CGS GHG299 are depressed and exhibit prominent striations that flare out towards the lateral margins of the bone (Figs. 6A, 6B, 6G and 6H) as described for *Scaloposaurus* (*Kemp, 1986*) and *Olivierosuchus* (*Fourie & Rubidge, 2007*; *Botha-Brink & Modesto, 2011*). The posterolateral margins of NMQR 48 and CGS GHG299 are weakly crenulated similar to those described for *Scaloposaurus* (*Kemp, 1986*), but contrasts with *Tetracynodon darti* (*Sigurdsen et al., 2012*) that exhibits prominent crenulations. As in *Promoschorhynchus*, the posterior margin of the sternum in NMQR 48, NMQR 3351, and CGS GHG299 does not exhibit a posterior notch, unlike those present in *Scaloposaurus* (*Kemp, 1986*), *Olivierosuchus* (*Fourie & Rubidge, 2007*; *Botha-Brink & Modesto, 2011*), and *Tetracynodon darti* (*Sigurdsen et al., 2012*).

NMQR 48, NMQR 3351, and CGS GHG299 all possess a prominent ventromedial ridge that originates at the posterior margin and extends anteriorly towards the centre of the bone, decreasing in height (Figs. 6A, 6B, 6G and 6H) as described for other therocephalians (*e.g.*, *Kemp, 1986*; *Fourie & Rubidge, 2007*; *Botha-Brink & Modesto, 2011*; *Huttenlocker, Sidor & Smith, 2011*; *Sigurdsen et al., 2012*). The ventromedial ridge is less pronounced in NMQR 48 compared to that of NMQR 3351 and CGS GHG299 and only extends to approximately halfway between the posterior margin and the centre of the bone. In contrast to NMQR 48, the ventromedial ridge extends to over a third of the length of the sternum in CGS GHG299 and to the centre of the sternum in NMQR 3351 (Table 2). As with the sternal shape, the condition of the ventromedial ridge is variable among therocephalian taxa. *Watson (1931*: 1176) stated that the sternum of *Ericiolacerta* is featureless, but he did state that it was incompletely exposed. *Kemp (1986)* described a weak ventromedial ridge for *Scaloposaurus*, but did not mention its extension on the sternum. A prominent ventromedial ridge that extends from the posterior margin to the centre of the bone is present in two specimens of *Olivierosuchus* (*Fourie & Rubidge, 2009*; *Botha-Brink & Modesto, 2011*) similar to that of the larger NMQR 3351, but this differs from the weak ventromedial ridge described for *Promoschorhynchus* (*Huttenlocker, Sidor & Smith, 2011*). The ventromedial ridge of *Tetracynodon* is distinctive as it runs along the entire length of the bone (*Sigurdsen et al., 2012*).

## Humerus

Humeri are preserved in NMQR 48, NMQR 3939, BP/1/4227, and NMQR 3351. The left humerus is preserved in NMQR 3939, but it is broken at the midshaft. The proximal end is only exposed in ventral view and is dorsoventrally compressed, the distal end is a separate element and still articulated to the proximal ends of the ulna and radius. The humeri of NMQR 48, BP/1/4227, and NMQR 3351 are all well-preserved and afford the best description (Fig. 7). NMQR 48 only preserves the proximal portion of the left humerus, however, the complete right humerus exists in the form of a cast and photographs of the original bone (Figs. 7A, 7C, 7E and 7G). BP/1/4227 preserves a complete left humerus, which is lying underneath the skull and is only visible in ventral view (Fig. 7F). Both humeri of NMQR 3351 are complete; the right humerus is exposed so that all surfaces, apart from the ventral surface of the proximal end, are visible (Figs. 7B, 7D and 7H). The left humerus is exposed so that only the proximal, dorsal, lateral surfaces, as well as the ventral surface of the proximal end can be observed.

The humerus is a robust bone with expanded proximal and distal ends that are connected by a short diaphysis, so that in dorsal and ventral view it is hourglass-shaped (Figs. 7A, 7B, 7E and 7F). The humeri of NMQR 48 and NMQR 3351 are slightly twisted so that the acute angle between the humeral head and the long axis of the distal end is approximately 35° similar to that described for *Scaloposaurus* (*Kemp, 1986*). The humeral torsion of therocephalian humeri have been reported as 33° for *Mirotenthes* (*Attridge, 1956*), 30° for *Olivierosuchus* (*Botha-Brink & Modesto, 2011*), and between 10°–25° for the humeri of early-diverging therocephalians (*Boonstra, 1964*). The well-expanded proximal and distal ends of NMQR 48, BP/1/4227, and NMQR 3351 are all sub-equal in width, each

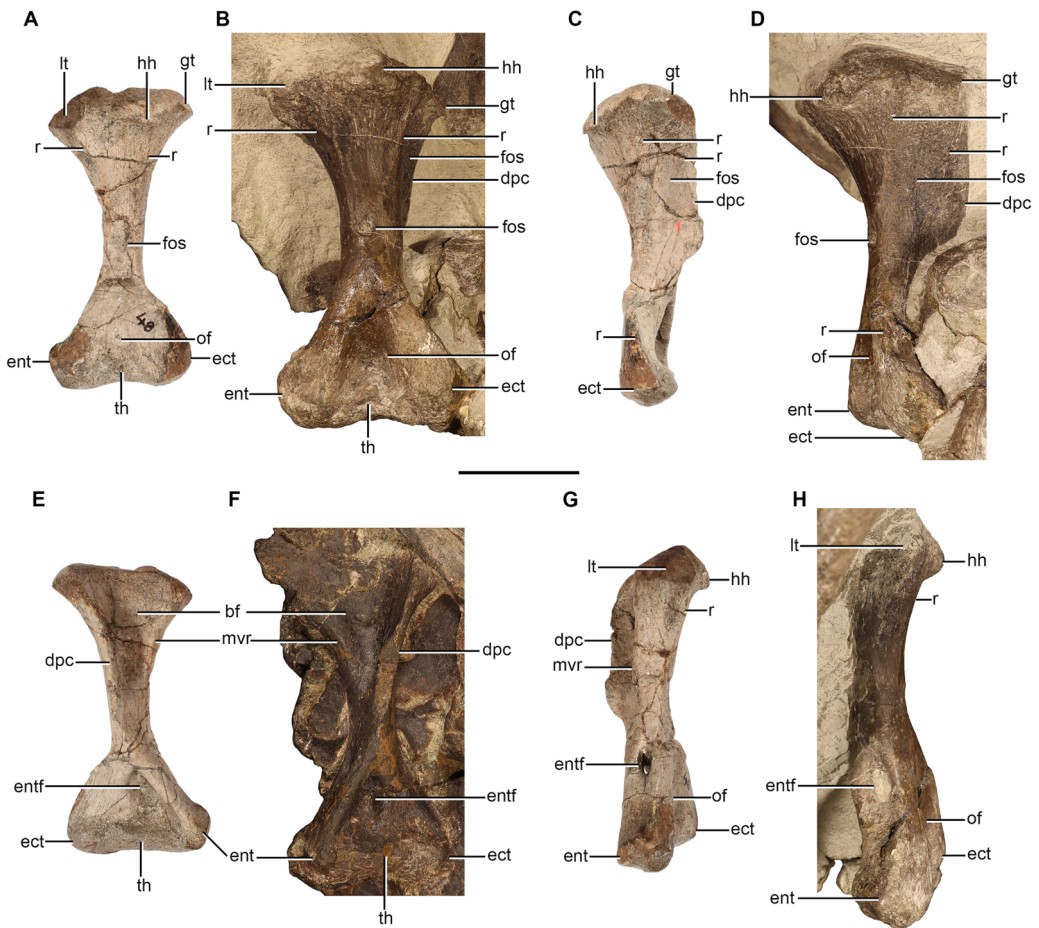

**Figure 7 Humeri of *Moschorhinus kitchingi*.** Right humerus of NMQR 48 (A) and NMQR 3351 (B) in dorsal view, right humerus of NMQR 48 (C) and NMQR 3351 (D) in lateral view, right humerus of NMQR 48 (E) and BP/1/4227 (F) in ventral view, and right humerus of NMQR 48 (G) and NMQR 3351 (H) in medial view. Scale bar equals 50 mm. Abbreviations: bf, bicipital fossa; dpc, deltopectoral crest; ect, ectepicondyle; ent, entepicondyle; entf, entepicondylar foramen; fos, fossa; gt, greater tuberosity; hh, humeral head; lt, lessor tuberosity; mvr, medioventral ridge; of, olecranon fossa; r, ridge; th, trochlea. Photographs by Brandon P. Stuart and Adam K. Huttenlocker.

measuring half the length of the bone (Table 3). This differs from the humeri described for *Mirotenthes* (*Attridge, 1956*) and *Tetracynodon darti* (*Sigurdsen et al., 2012*) in which the proximal and distal ends are not extensively wide, measuring a third of the length of the humerus and a quarter respectively. The proximal surfaces of NMQR 48 and NMQR 3351 are convex and rugose. The proximodorsal margin is strongly mediolaterally convex and conversely the proximoventral margin is strongly mediolaterally concave.

In dorsal view the proximal end of the bone is divided into three surfaces: a lateral, dorsal, and medial surface (Figs. 7A and 7B). The dorsal surface is approximately triangular in outline with the greatest mediolateral expansion at the proximal end. The triangular dorsal surface recedes mediolaterally distally to the middle of the diaphysis where a conspicuous oval fossa is present (Figs. 7A and 7B), which is particularly deep in NMQR 3351. The humeral head is dorsally inflected and is positioned on the

**Table 3 Measurements of the humeri, radii, and ulnae of specimens of *Moschorhinus kitchingi*.**

| | NMQR 48 | | NMQR 3939 | BP/1/4227 | NMQR 3351 | |
| --- | --- | --- | --- | --- | --- | --- |
| | Left | Right | Left | Left | Left | Right |
| **Humerus** | | | | | | |
| Length | – | 111.78 | – | 113.39* | 149.28* | 153.73 |
| Width of proximal end | 56.83* | 55.40 | 65.86* | 63.63* | – | 81.90* |
| Width of distal end | – | 54.67 | 53.62* | 62.17* | 71.82* | 79.43* |
| Width of shaft | – | 15.07 | – | 18.18* | 24.52 | 25.34 |
| Length of deltopectoral crest | – | 62.35 | 73.11* | 61.57* | – | 73.98* |
| **Radius** | | | | | | |
| Length | – | – | 61.93* | 99.65 | 100.34 | |
| Width of proximal end | – | – | 31.21* | 29.56 | 40.92 | |
| Width of distal end | – | – | 28.28* | 28.15 | 37.30 | |
| **Ulna** | | | | | | |
| Length | – | – | 84.97* | 109.27* | – | |
| Width of proximal end | – | – | 38.31* | 37.95 | – | |
| Width of distal end | – | – | – | 16.55 | – | |

Notes:
All measurements are in mm.
* Indicates element is partially covered by matrix or incomplete.
Measurements of the radius and ulna of BP/1/4227 before sampling for thin sectioning were provided by Adam Huttenlocker.

proximodorsal margin, slightly closer to the lateral margin than the medial (Figs. 7A–7D, 7G and 7H). The humeral head is most prominent at its midpoint and recedes in height laterally and medially until the lateral and medial ends of the proximodorsal margins are raised by the greater and lesser tuberosity respectively, with the former being slightly more developed than the latter (Figs. 7A and 7B).

The lateral and dorsal surfaces are delimited by a low ridge, the anterior dorsoventral line (ADVL *Sensu Boonstra, 1964*). The ridge extends distally from the proximodorsal margin of the proximal end, receding in height, to approximately the centre of the diaphysis (Figs. 7A–7D). Lateral to this ridge, an anteroposteriorly long and ventrally extended lamina forms the lateral surface of the deltopectoral crest (Figs. 7C and 7D). This surface is weakly depressed at its centre and raised ventrally by a second low ridge that extends distally from the greater tuberosity (Figs. 7C and 7D). NMQR 48, BP/1/4227, and NMQR 3351 all exhibit a well-developed posteriorly and ventrally expanded deltopectoral crest (Figs. 7C–7F) in contrast to the weak deltopectoral crest described for the humeri of early-diverging therocephalians (*Boonstra, 1964*). The deltopectoral crest of NMQR 48 extends posteriorly to just over half the length of the bone and to just under half the length of the bone in BP/1/4227 and NMQR 3351 (Table 3). The distal end of the deltopectoral crest of NMQR 48 and NMQR 3351 sharply recedes to terminate smoothly into the shaft where it continues as a rounded rod-like flange on the ventral surface that extends distomedially to the ventral margin of the entepicondyle (Figs. 7C–7F). The ventral surface of the proximal ends of NMQR 48 and BP/1/4227 are dominated by a deeply mediolaterally concave triangular bicipital fossa (Figs. 7E and 7F). The bicipital fossa is

delimited laterally by the medial surface of the deltopectoral crest and medially by a broad, rounded medioventral ridge that extends distally from the proximoventral margin (Figs. 7E and 7F).

The medial and dorsal surfaces of the proximal end are delimited by a sharp and short ridge, the lateromedial line (LML *Sensu Boonstra, 1964*). The ridge extends distally from the lesser tuberosity to approximately a third of the way from the middle of the shaft (Figs. 7A, 7B, 7G and 7H). The medial surface of the proximal end is smooth, dorsoventrally convex, and is delimited ventrally by the medioventral ridge described above. A large oval entepicondylar foramen is present distally on the medial surface of NMQR 48, BP/1/4227, and NMQR 3351 (Figs. 7E–7H) as in other eutherocephalians (*Kemp, 1986*; *Fourie & Rubidge, 2007*; *Botha-Brink & Modesto, 2011*; *Huttenlocker, Sidor & Smith, 2011*; *Sigurdsen et al., 2012*), apart from *Mirotenthes* of which *Attridge (1956*: 84) states that there is no trace of an entepicondylar foramen (although this could be due to lack of preparation in that specimen). The entepicondylar foramen is enclosed ventrally by the rod-like flange described above and opens into the ventral surface of the distal end, which bears a deep trough (Figs. 7E and 7F). There is no ectepicondylar foramen as in all other eutherocephalians (*Kemp, 1986*; *Fourie & Rubidge, 2007*; *Botha-Brink & Modesto, 2011*; *Huttenlocker, Sidor & Smith, 2011*; *Sigurdsen et al., 2012*) in contrast to that described for early-diverging therocephalians (*Boonstra, 1964*). Lateral to the trough, the surface is slightly raised and then becomes inflected, forming a sharp ridge on the lateral surface that extends from the distal margin of the ectepicondyle to almost the middle of the shaft (Figs. 7C–7F). The epicondyles of NMQR 48 and NMQR 3351 are approximately triangular in outline in dorsal view and bear a deep triangular olecranon fossa, which separates the entepicondyle and ectepicondyle (Figs. 7A and 7B). The entepicondyles of NMQR 48 and NMQR 3351 bear a conspicuous hook-like projection, which is best seen in dorsal view (Figs. 7A and 7B).

The trochlea is present on the distal surface of NMQR 48 and NMQR 3351 and separates the entepicondyle and ectepicondyle. The trochlea extends on both the dorsal and ventral surface of the epicondyle (Figs. 7A, 7B, 7E and 7F). The trochlea forms a bulbous protuberance on the distodorsal margin of the olecranon fossa, which is far more developed in NMQR 3351 than in NMQR 48 (Figs. 7A and 7B). On the distoventral margin of NMQR 48 and BP/1/4227 the trochlea forms a comparatively mediolaterally wider, but less pronounced protuberance that gradually recedes in height laterally and medially (Figs. 7A and 7B).

### Radius

Radii are preserved in NMQR 48, NMQR 3939, BP/1/4227, and NMQR 3351. The distal end of the right radius is preserved in NMQR 48, but it is not well preserved and largely covered by matrix. The complete right radius is preserved in NMQR 3351, but it is lying underneath the humerus and only the proximal and distal articulatory surface, along with a small section of the anterior surface of the shaft are exposed (Figs. 8A–8C). The proximal end of the left radius of NMQR 3939 is still articulated with the proximal end of the ulna and distal end of the humerus (Figs. 8D–8G) and the distal end is preserved on the ventral

side of the skull. A cast of the left forelimb elements of NMQR 3939 exists which includes portions of the mid-shafts of the ulna and the radius (Figs. 8H and 8I). The left radius is preserved in BP/1/4227 and is still articulated with the ulna, but is preserved in two pieces with a missing portion of the shaft (Figs. 8J–8M).

The radius is a short and relatively slender bone with expanded proximal and distal ends that are separated by a circular and narrow shaft. The radii of NMQR 3351, NMQR 3939, and BP/1/4227 are relatively straight, but medially inflected at their distal ends (Figs. 8A, 8I–8L). The radius of NMQR 3351 is approximately 65% of the length of the humerus differing from BP/1/4227 in which the radius is approximately 75% the length of the humerus. The proximal end of NMQR 3351 is approximately rectangular in outline and the articulatory surface is broad and shallowly concave (Fig. 8B) similar to that described for other therocephalians (*Kemp, 1986*; *Fourie & Rubidge, 2007*; *2009*; *Botha-Brink & Modesto, 2011*; *Huttenlocker, Sidor & Smith, 2011*). The distal end is approximately oval in outline and the articulatory surface is flat (Fig. 8C). The proximal end is more expanded than the distal end in contrast to that described for *Promoschorhynchus* (*Huttenlocker, Sidor & Smith, 2011*). The lateral and medial surfaces of the radius of NMQR 3351 and NMQR 3939 are delimited by a sharp longitudinal ridge on the posterior margin of the bone (Figs. 8A, 8D and 8F) as described for other therocephalians (*Kemp, 1986*; *Fourie & Rubidge, 2007*; *Huttenlocker, Sidor & Smith, 2011*; *Liu & Abdala, 2019*).

## Ulna

Ulnae are preserved in NMQR 48, NMQR 3939, BP/1/4227, and NMQR 3351. The distal end of the right ulna of NMQR 48 is preserved, but as with the radius it is obscured by matrix. Only a small section of the distal end of the right ulna is exposed in NMQR 3351. The left ulna of BP/1/4227 is relatively well preserved, but is missing a section of the midshaft (Figs. 8J–8M). The proximal end of the left ulna of NMQR 3939 is preserved (Figs. 8D and 8E) and the proximal end and a portion of the midshaft are represented in the cast of the forelimb (Figs. 8H and 8I).

The proximal surface of NMQR 3939 is largely covered by matrix and by the distal end of the humerus (Figs. 8D–8G) and is damaged in BP/1/4227 (Figs. 8J–8M). From what can be observed, the proximal articulation facet is smooth and shallow (Fig. 8F). The proximal surface is slightly raised towards the olecranon region, which bears an ossified but poorly-developed olecranon process (Figs. 8D–8G) differing from the short and broad olecranon process described for early-diverging therocephalians (*Boonstra, 1964*; *Fourie & Rubidge, 2009*; *Abdala et al., 2014b*). The olecranon region of BP/1/4227 superficially appears to be more developed, but this is exaggerated due to the damaged proximal articulatory surface (Figs. 8J–8M). The ossified but poorly-developed olecranon process of NMQR 3939 and BP/1/4227 is generally consistent with those described for the eutherocephalians *Olivierosuchus* (*Fourie & Rubidge, 2007*; *Botha-Brink & Modesto, 2011*), *Scaloposaurus* (*Huttenlocker et al., 2022*), and *Microgomphodon* (*King, 1996*). The absence of an ossified olecranon process is commonly reported for small therocephalian specimens that are inferred to be skeletally immature such as those of *Scaloposaurus* (*Kemp, 1986*) and *Tetracynodon darti* (*Sigurdsen et al., 2012*), but has also been reported as being absent

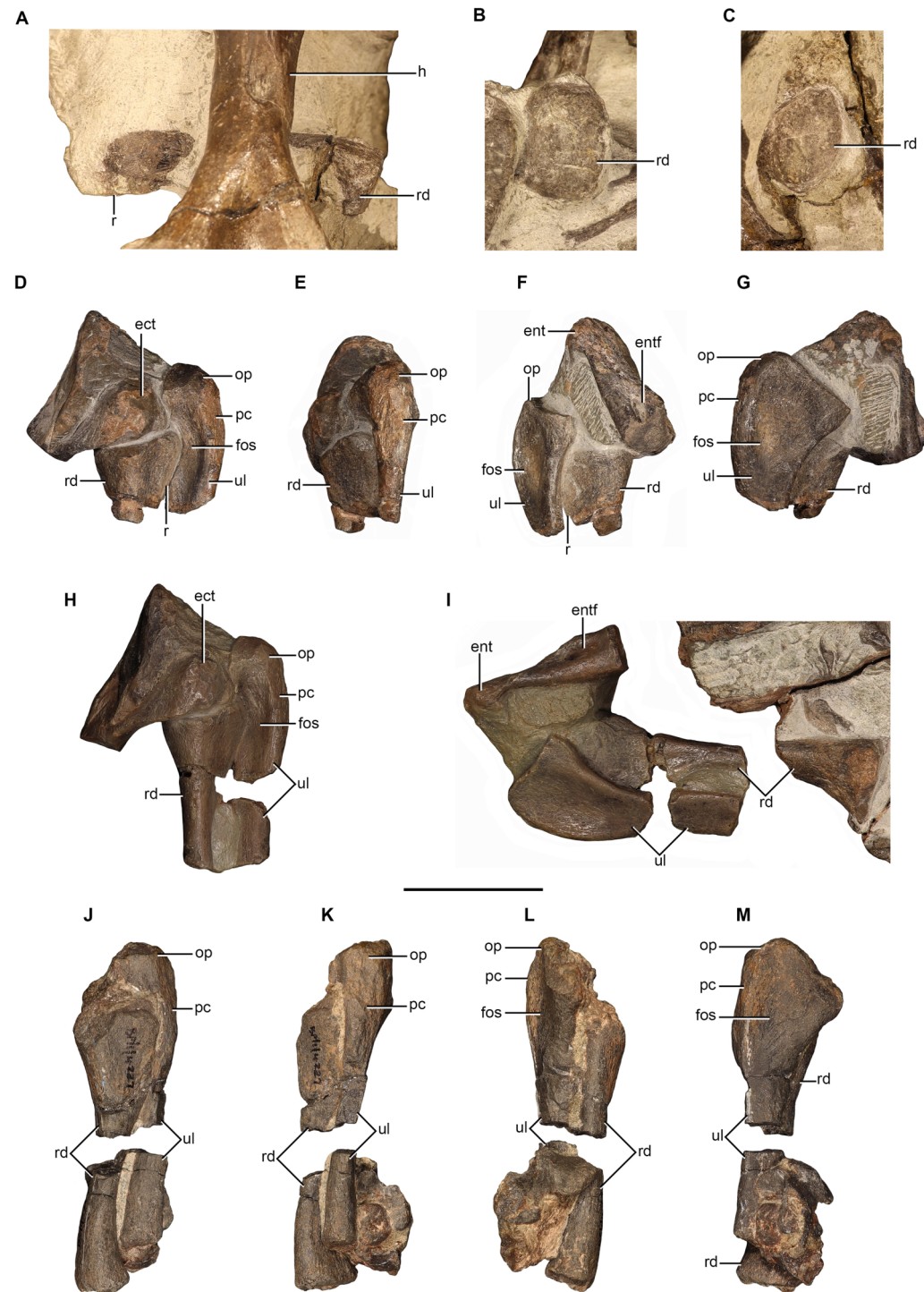

**Figure 8 Radii and ulnae of *Moschorhinus kitchingi*.** Right radius of NMQR 3351 in medial (A), proximal (B), and distal (C) views, left radius and ulna of NMQR 3939 in lateral (D), posterior (E), anterior (F), and medial (G) views, cast of left radius and ulna of NMQR 3939 in lateral (H) and medial (I) views, left radius and ulna of BP/1/4227 in lateral (J), posterior (K), anterior (L), and medial (M) views. Scale bars equal 50 mm. Abbreviations: ect, ectepicondyle; ent, entepicondyle; entf, entepicondyle foramen; fos, fossa; h, humerus; op, olecranon process; pc, posterior crest; r, ridge; rd, radius; ul, ulna. Photographs by Brandon P. Stuart.

in the large scylacosaurid described by *Cys (1967)* and the large akidnognathid *Jiufengia* (*Liu & Abdala, 2019*).

The lateral surface of the proximal end of BP/1/4227 is completely obscured by the proximal end of the radius (Fig. 8J), but the posterior portion of the lateral surface of NMQR 3939 is visible (Fig. 8D). A deep fossa that extends distally is present as described for *Simorhinella* (*Abdala et al., 2014b*), but the distal extension of the fossa is uncertain due to the missing sections of NMQR 3939. The fossa does not appear to be present on the section of the shaft represented on the cast (Fig. 8H) potentially differing from the ulna of *Simorhinella* where the fossa is followed distally by an anterior lineation (*Abdala et al., 2014b*). A mediolaterally broad posterior crest is present proximally on the posterior margin of NMQR 3939 and BP/1/4227 (Figs. 8D, 8E, 8G, 8J, 8K, 8L and 8M). The posterior crest becomes reduced distally and borders the fossa on the lateral surface posteriorly as described for *Scaloposaurus* (*Kemp, 1986*) and *Simorhinella* (*Abdala et al., 2014b*). The medial surface of NMQR 3939 and BP/1/4227 are dominated by an anteroposteriorly long and shallow fossa that attenuates distally and is bordered by the posterior crest (Figs. 8F, 8G, 8L and 8M) as described for all other therocephalians in which the ulna is known (*e.g.*, *Kemp, 1986*; *King, 1996*; *Botha-Brink & Modesto, 2011*; *Abdala et al., 2014b*).

## Manus

Manual elements are preserved in NMQR 48, NMQR 3939, BP/1/4227, and NMQR 3351. NMQR 48 preserves a few carpal elements, but they are largely covered by matrix. NMQR 3939 preserves the left manus in ventral view and it is the most complete of all the studied material (Figs. 9A and 9B). BP/1/4227 preserves three metacarpals in ventral view and possible carpal elements, but they are too poorly preserved to identify (Figs. 9C and 9D). NMQR 3351 preserves the right manus in dorsal view and consists of three metacarpals (possibly four, see below), carpal, and phalangeal elements (Figs. 9E and 9F).

### Carpus

No pisiform or intermedium are preserved in any of the studied material. The radiale of NMQR 3939 and NMQR 3351 is quadrangular in shape and is the largest bone in the carpus (Figs. 9A, 9B, 9E and 9F) similar to that of *Olivierosuchus* (*Fourie & Rubidge, 2007*; *Botha-Brink & Modesto, 2011*), and *Microgomphodon* (*Abdala et al., 2014a*). The quadrangular shape differs from the square radiale described for *Tetracynodon darti* (*Fontanarossa et al., 2018*). The radiale is mediolaterally broad and proximodistally short and two depressions are present: a proximal concavity on the ventral surface of NMQR 3939 (Figs. 9A and 9B) and a distolateral concavity on the dorsal surface of NMQR 3351 (Figs. 9E and 9F). The proximal margin is slightly convex in NMQR 3351 (Figs. 9E and 9F), but less so in NMQR 3939, although this is attributed to a crack through the proximal margin (Figs. 9A and 9B). The distomedial margin is strongly concave and the distolateral margin is slightly convex (Figs. 9A, 9B, 9E, and 9F).

The only specimen that preserves an ulnare is NMQR 3939 (Figs. 9A and 9B). The ulnare is proximodistally expanded and mediolaterally narrow making it the longest bone in the carpus (Table 4). The rectangular shape is similar to that described for

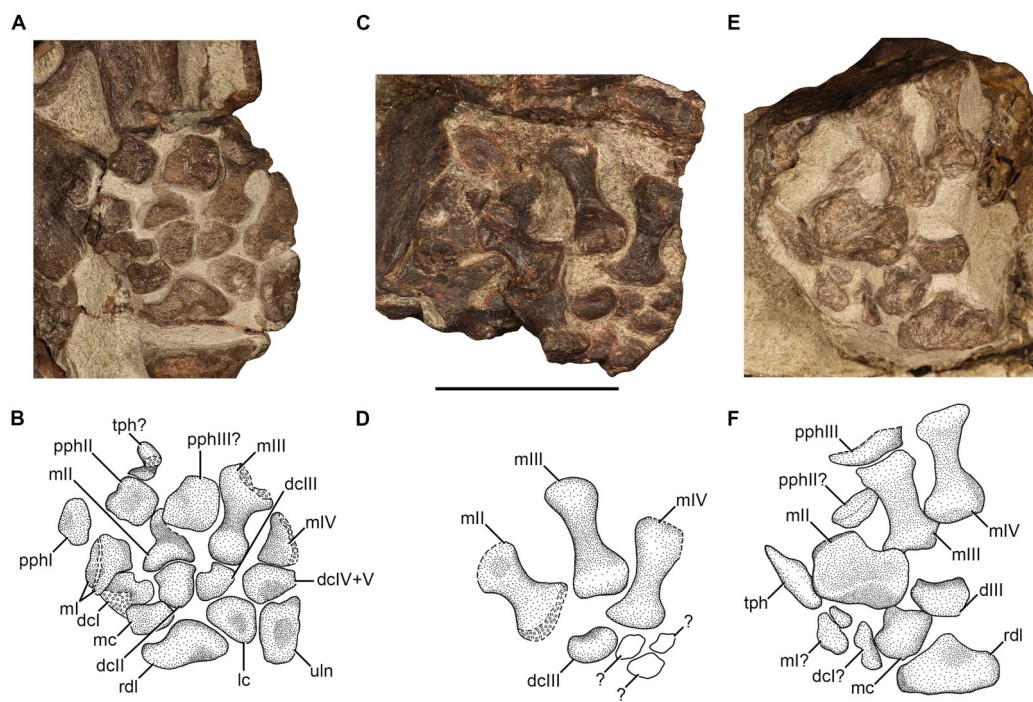

**Figure 9 Manus of *Moschorhinus kitchingi*.** Photographs and stipple drawings of the left manus of NMQR 3939 in ventral view (A and B), left manus in BP/1/4227 ventral view (C and D), and right manus of NMQR 3351 in dorsal view (E and F). Hatching indicates damaged surfaces. Scale bar equals 50 mm. Abbreviations: dc, distal carpal; lc, lateral centrale; m, metacarpal; pph proximal phalanx; rdl, radiale; tph, terminal phalanx; uln, ulnare. Photographs and illustrations by Brandon P. Stuart.

*Olivierosuchus* (*Fourie & Rubidge, 2007*) and *Tetracynodon darti* (*Fontanarossa et al., 2018*), but differs from the hourglass-shaped ulnare described for early-diverging therocephalians (*Cys, 1967*; *Fourie & Rubidge, 2009*; *Kümmel et al., 2020*) and *Ictidosuchoides* (*Fourie, 2013*; *Kümmel et al., 2020*). A circular depression is present centrally on the ventral surface (Figs. 9A and 9B).

The medial centrale of NMQR 3939 and NMQR 3351 is approximately rectangular in shape (Figs. 9A, 9B, 9E, and 9F) similar to that described for early-diverging therocephalians (*Fourie & Rubidge, 2009*), but differs from the oval medial centrale of *Tetracynodon darti* (*Kümmel et al., 2020*). The proximal margin is slightly convex to articulate with the concave distomedial margin of the radiale and the distal medial margin is strongly convex (Figs. 9A, 9B, 9E, and 9F). NMQR 3939 is the only specimen that preserves a lateral centrale (Figs. 9A and 9B). The lateral centrale is irregularly oval in shape similar to that of *Olivierosuchus* (*Fourie & Rubidge, 2007*; *Botha-Brink & Modesto, 2011*; *Kümmel et al., 2020*), and *Tetracynodon darti* (*Kümmel et al., 2020*), but differs from the early-diverging therocephalian described by *Fourie & Rubidge (2009)*, which has a proximally narrow lateral centrale that is mediolaterally expanded distally (*Kümmel et al., 2020*). The medial margin is weakly concave to articulate with the convex distolateral margin of the radiale and the lateral margin is straight to articulate with the straight medial margin of the ulnare (Figs. 9A and 9B).

**Table 4 Measurements of the manual elements of specimens of *Moschorhinus kitchingi*.**

| | NMQR 3939 | BP/1/4227 | NMQR 3351 |
|---|---|---|---|
| | Left | Left | Right |
| **Radiale** | | | |
| Length | 12.46* | – | 16.61 |
| Width | 22.46 | – | 28.33 |
| **Ulnare** | | | |
| Length | 18.75 | – | – |
| Width | 13.05 | – | – |
| **Medial centrale** | | | |
| Length | 7.75 | – | 11.78 |
| Width | 12.04 | – | 15.25 |
| **Lateral centrale** | | | |
| Length | 10.18 | – | – |
| Width | 14.98 | – | – |
| **Distal carpal I** | | | |
| Length | 10.17* | – | – |
| Width | 12.12* | – | – |
| **Distal carpal II** | | | |
| Length | 11.16 | – | – |
| Width | 8.91 | – | – |
| **Distal carpal III** | | | |
| Length | 7.28 | 7.36 | 9.48 |
| Width | 8.91 | 11.95 | 15.90 |
| **Distal carpal IV+V** | | | |
| Length | 14.90 | – | – |
| Width | 11.51 | – | – |
| **Metacarpal I** | | | |
| Length | 8.51 | – | – |
| Width | 15.86 | – | – |
| **Metacarpal II** | | | |
| Length | 14.24* | 25.31* | 27.02 |
| Width of proximal end | 13.05 | 15.38* | 14.92 |
| Width of midshaft | 6.48* | 9.13 | 12.73 |
| Width of distal end | – | 15.95* | 19.04 |
| **Metacarpal III** | | | |
| Length | 22.35* | 29.99 | 28.42 |
| Width of proximal end | 10.61 | 13.43 | 16.11 |
| Width of midshaft | 6.25 | 6.35 | 9.08 |
| Width of distal end | 16.06* | 12.95 | 14.46 |
| **Metacarpal IV** | | | |
| Length | 11.14* | 28.13* | 31.20 |
| Width of proximal end | – | 14.88* | 17.57 |
| Width of midshaft | – | 7.87* | 7.39 |
| Width of distal end | – | 11.11 | 14.81 |

| Table 4 (continued) | NMQR 3939 | BP/1/4227 | NMQR 3351 |
|---|---|---|---|
| | Left | Left | Right |
| **Proximal phalanx I** | | | |
| Length | 8.89 | – | – |
| Width | 11.46 | – | – |
| **Proximal phalanx II** | | | |
| Length | 11.53 | – | – |
| Width | 11.24 | – | – |
| **Proximal phalanx III** | | | |
| Length | 12.66? | – | – |
| Width | 13.67? | – | 20.98 |
| **Terminal phalanx I** | | | |
| Length | – | – | 19.98 |

**Notes:**
All measurements are in mm.
* Indicates element is partially covered by matrix or incomplete.
? Indicates identification is uncertain.

NMQR 3939 preserves all the distal carpals (Figs. 9A and 9B). BP/1/4227 (Figs. 9C and 9D) and NMQR 3351 (Figs. 9E and 9F) only preserves distal carpal III, identified based on the comparison with NMQR 3939 and position, along with fragments of bone that could represent distal carpals, but cannot be identified with any confidence. The distal carpals of NMQR 3939 (Figs. 9A and B) are all approximately similar in size, but distal carpal IV+V is slightly larger than the rest similar to those of early-diverging therocephalians (*Boonstra, 1964*: fig 4; *Cys, 1967*; *Fourie & Rubidge, 2009*; *Kümmel et al., 2020*: fig. 6B). Distal carpal III is only slightly smaller than the rest as in *Olivierosuchus* (*Botha-Brink & Modesto, 2011*) and *Tetracynodon darti* (*Fontanarossa et al., 2018*; *Kümmel et al., 2020*). The relative size of distal carpal I in comparison with the rest of the distal carpals of NMQR 3939 differs from most eutherocephalians in which distal carpal I is usually larger than the rest (*e.g.*, *Attridge, 1956*; *Botha-Brink & Modesto, 2011*; *Fourie, 2013*; *Fontanarossa et al., 2018*; *Kümmel et al., 2020*).

The medial side of the ventral surface of distal carpal I of NMQR 3939 is damaged (Figs. 9A and 9B), but the outline of the bone can be seen as square as in *Olivierosuchus* (*Botha-Brink & Modesto, 2011*; *Kümmel et al., 2020*), and *Ictidosuchoides* (*Fourie, 2013*; *Kümmel et al., 2020*) differing from *Tetracynodon darti* (*Fontanarossa et al., 2018*) and *Microgomphodon* (*Abdala et al., 2014a*) in which it is rectangular. The lateral margin is deeply concave as in *Olivierosuchus* (*Botha-Brink & Modesto, 2011*: fig 7). The bone fragments lying medial to the medial centrale of NMQR 3351 could potentially be distal carpal I due to the concave surface resembling that of distal carpal I of NMQR 3939 (Figs. 9E and 9F).

Distal carpal II of NMQR 3939 is quadrangular similar to other therocephalians (*e.g.*, *Fourie & Rubidge, 2007*, *2009*; *Fontanarossa et al., 2018*) and has a convex distal margin to articulate with metacarpal II. Distal carpal III of NMQR 3939, BP/1/4227, and NMQR

3351 is oval as in other therocephalians (*e.g.*, *Botha-Brink & Modesto, 2011*; *Kümmel et al., 2020*) and has a deeply concave medial margin (Figs. 9A–9F). Distal carpals IV and V are fused in NMQR 3939 (Figs. 9A and 9B) which is common in therocephalians (*e.g.*, *Attridge, 1956*; *Cys, 1967*; *Hopson, 1995*; *Fourie & Rubidge, 2007*, *2009*; *Botha-Brink & Modesto, 2011*; *Kümmel et al., 2020*), although there are separate ossifications in *Ictidosuchoides* (*Fourie, 2013*; *Kümmel et al., 2020*), and distal carpal V is inferred to have been cartilaginous in *Tetracynodon darti* (*Fontanarossa et al., 2018*). No fusion line is visible on distal carpal IV and V of NMQR 3939, but according to *Kümmel et al. (2020)* fusion lines may not always be present. Distal carpal IV and V is oval in shape similar to other therocephalians (*e.g.*, *Attridge, 1956*; *Cys, 1967*; *Botha-Brink & Modesto, 2011*; *Kümmel et al., 2020*) and a depression is present on the proximal margin (Figs. 9A and 9B).

### Metacarpus

NMQR 3351 preserves small pieces of bones distal to the medial centrale that could represent metacarpal I, but cannot be confirmed with confidence (Figs. 9E and 9F). A crack runs through metacarpal I of NMQR 3939, but the overall shape of the bone is relatively undisturbed (Figs. 9A and 9B). Metacarpal I of NMQR 3939 is proximodistally short and mediolaterally broad making the overall shape rectangular similar to that of *Jiufengia* (*Liu & Abdala, 2019*) but differs from the quadrangular, slightly waisted, metacarpal I of *Olivierosuchus* (*Botha-Brink & Modesto, 2011*) and other therocephalians in which metacarpal I is known (*e.g.*, *Fourie & Rubidge, 2007*; *Fontanarossa et al., 2018*). The proximal end of metacarpal II of NMQR 3939 is exposed but the majority of the distal end is lying underneath a bone tentatively identified as proximal phalanx III (Figs. 9A and 9B). Metacarpal II of NMQR 3939 is mediolaterally expanded proximally, slightly waisted at its shaft, and proximodistally longer than metacarpal I but shorter than metacarpal III. The metacarpals of BP/1/4227 and NMQR 3351, identified as metacarpal II based on their resemblance to metacarpal II of NMQR 3939, are both proximodistally shorter and mediolaterally broader than the more lateral metacarpals (Figs. 9C–9F). This is particularly true for metacarpal II of NMQR 3351, which strongly resembles the mediolaterally broad metacarpal II of *Jiufengia* (*Liu & Abdala, 2019*). Metacarpal III of NMQR 3939, BP/1/4227, and NMQR 3351 and metacarpal IV of BP/1/4227 and NMQR 3351 are all proximodistally long with mediolaterally expanded proximal and distal ends and waisted shafts (Figs. 9A–9F). Metacarpal IV of NMQR 3939 is broken and only consists of a fragment of the proximal end. Metacarpal IV of NMQR 3351 is the longest metacarpal but is only slightly longer than metacarpal III. Metacarpal III of BP/1/4227 is slightly longer than metacarpal IV but this is here attributed to the damaged distal end of metacarpal IV (Figs. 9C and 9D).

### Phalanges

The bone lying distally to metacarpal I of NMQR 3939 is identified as proximal phalanx I and the bone lying distomedially to metacarpal II is identified as proximal phalanx II (Figs. 9A and 9B). Proximal phalanx I is oval in shape and is smaller than the square proximal phalanx II (Figs. 9A and 9B). A large square bone that is overlying metacarpal II of NMQR 3939 could represent proximal phalanx III, but its position does not permit a positive

identification as it is lying medially to metacarpal III. A small oval shaped bone is lying distolaterally to metacarpal II in NMQR 3351 could represent proximal phalanx II (Figs. 9E and 9F). A broken fragment of bone lying distal to metacarpal III of NMQR 3351 is identified as the remains of proximal phalanx III due to its position (Figs. 9E and 9F). A small, rounded bone fragment lying distally to proximal phalanx II of NMQR 3939 could represent a terminal phalanx, but the incompleteness of the distal portion of the manus does not allow for a positive identification (Figs. 9A and 9B). A bone lying distally to metacarpal II of NMQR 3351 is identified as a potential terminal phalanx I (Figs. 9E and 9F). The terminal phalanx is mediolaterally narrow and proximodistally long and attenuates to a rounded point.

## Ilium

Ilia are preserved in NMQR 3939, SAM-PK-K10698, and NMQR 3351. NMQR 3351 preserves both complete ilia (Figs. 10A–10C). The lateral surface of the right ilium is exposed but has suffered from slight dorsoventral compression and is damaged along its anterior margin. The proximal head of the right femur is still in articulation with its acetabular facet (Fig. 10B). The lateral surface of the left ilium as well as the acetabular facet is exposed (Fig. 10A and 10C). The left ilium of NMQR 3939 is incomplete and is only represented by the majority of the iliac acetabular contribution, which is still articulated to the rest of the acetabulum (Fig. 10D). SAM-PK-K10698 preserves a complete right ilium that is only exposed in medial view (Figs. 10E and 10F). Measurements of the ilia are given in Table 5.

The ilium is comprised of a mediolaterally thin and anteroposteriorly expanded blade with a ventrally situated iliac acetabular facet. The anterior margin of the ilium possesses two distinct processes, an anterodorsal and an anteroventral process (Figs. 10A–10C, 10E and 10F), a characteristic of all therocephalian taxa in which the ilium is known (*Kemp, 1978*, *1986*; *Fourie & Rubidge, 2007*; *Sidor et al., 2014*). However, the two anterior processes of the iliac blade are less distinctive in early-diverging therocephalians (*Boonstra, 1964*; *Fourie & Rubidge, 2009*). The two processes are separated by a shallow triangular fossa on the lateral surface of the blade (Figs. 10A–10C) as described for *Regisaurus* (*Kemp, 1978*) and *Olivierosuchus* (*Fourie & Rubidge, 2007*). The anteroventral process of the left ilium of NMQR 3351 and SAM-PK-K10698 is short and blunt compared to the anteroventral process of the right ilium of NMQR 3351 (Figs. 10A, 10B, 10E and 10F). This is attributed to the compression and the damage to the right ilium of NMQR 3351. The short and blunt anteroventral process differs from the anteriorly extended and dorsoventrally narrow finger-like process described for *Regisaurus* (*Kemp, 1978*), *Scaloposaurus* (*Kemp, 1986*), and *Olivierosuchus* (*Fourie & Rubidge, 2007*). The anterodorsal process terminates on the dorsal margin of the blade as a broad rounded depression. Broad, low ridges extend from the distal margin of both of the anterior processes to the centre of the iliac blade where they converge dorsally just above the supraacetabular buttress. The lateral surface of the posterior half of the blade of NMQR 3351 is only slightly dorsoventrally concave with the dorsal and ventral margins of the blade attenuating in the posterior process (Figs. 10A and 10B).

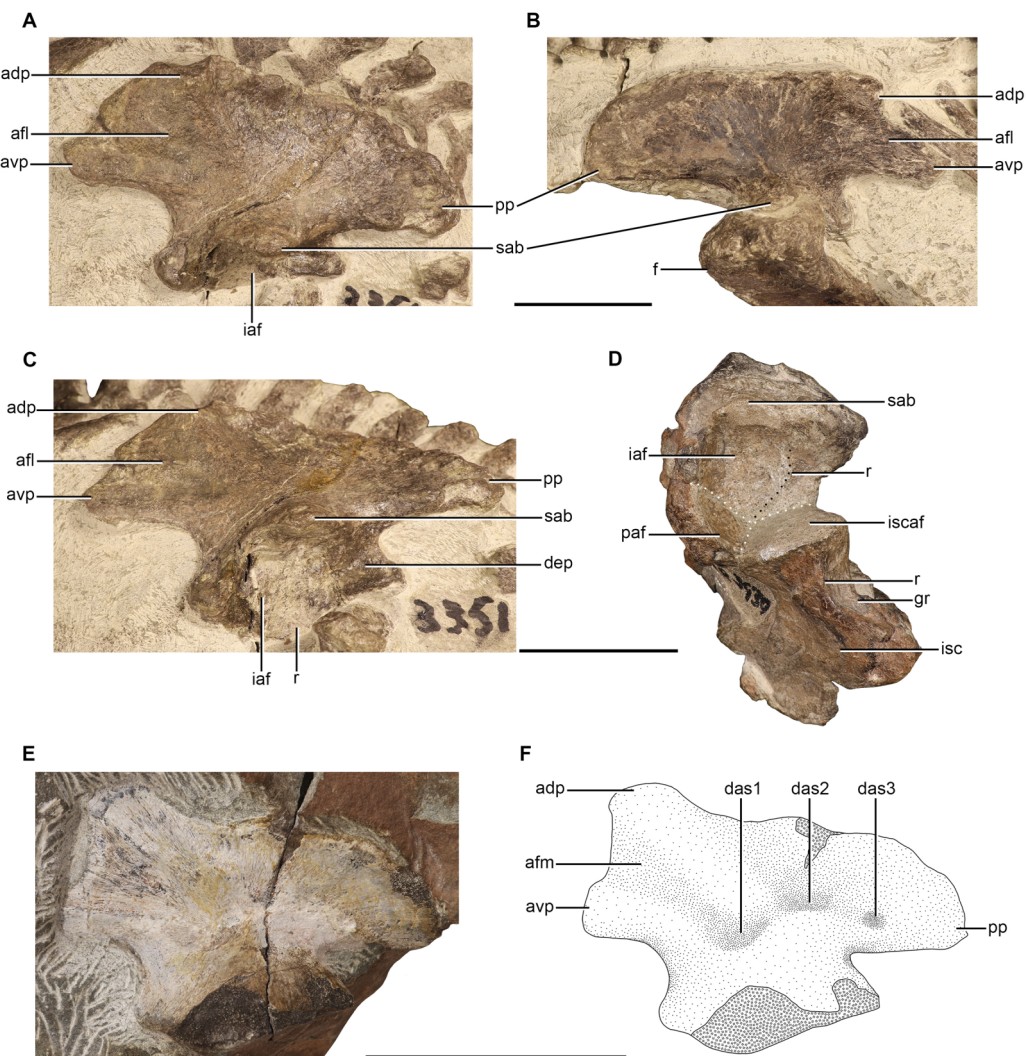

**Figure 10  Ilia of *Moschorhinus kitchingi*.** Left (A) and right (B) ilium of NMQR 3351 in lateral view, left ilium of NMQR 3351 in ventrolateral view (C), left pelvic girdle of NMQR 3939 in ventral view (D), photograph (E) and stipple drawing (F) of the right ilium of SAM-PK-K10698 in medial view. Hatching indicates damaged surfaces. Scale bars all equal 50 mm. Abbreviations: adp, anterodorsal process of the ilium; afl, anterior fossa on the lateral surface of the ilium; afm, anterior fossa on the medial surface of the ilium; avp, anteroventral process of the ilium; das, depression for the attachment of sacral rib; dep, depression; f, femur; gr, groove; iaf, acetabular facet of the ilium; isc, ischium; iscaf, acetabular facet of the ischium; pp, posterior process of the ilium; r, ridge; sab, supraacetabular buttress of the ilium. Photographs and illustration by Brandon P. Stuart.               

The ventral base of the iliac blade is thickened through the laterally expanded supraacetabular buttress, which is present at the midline of the ilium (Figs. 10A–10D). The supraacetabular buttress extends laterally, tapering and terminating in a rounded end, which contributes to much of the dorsal surface of the iliac acetabular facet. On the anterior portion of the supraacetabular buttress, the dorsal and anterior margin of the iliac acetabular facet is deeply medially concave that sharply delimits the anterior internal surface of the iliac acetabular facet from the anteroventral edge of the ilium. Posterior to

Table 5 Measurements of the ilia, pubis, and ischia of specimens of *Moschorhinus kitchingi*.

| | SAM-PK-K10698 | NMQR 48 | | NMQR 3568 | NMQR 3939 | NMQR 3351 | |
|---|---|---|---|---|---|---|---|
| | Right | Left | Right | Right | Left | Left | Right |
| **Ilium** | | | | | | | |
| Length of blade | 75.03 | – | – | – | – | 134.02* | 130.31* |
| Length of acetabular facet | 42.89 | – | – | – | 57.53* | 63.85 | – |
| **Pubis** | | | | | | | |
| Length of blade | – | – | 57.30 | 65.52 | – | – | – |
| Length of acetabular facet | – | – | 19.97* | 33.59 | 30.27* | – | – |
| **Ischium** | | | | | | | |
| Length of blade | – | 42.72* | – | – | 63.53* | – | – |
| Length of acetabular facet | – | 32.09* | – | – | 34.59* | – | 39.34* |

Notes:
All measurements are in mm.
\* Indicates element is partially covered by matrix or incomplete

the supraacetabular buttress, the dorsal margin of the iliac acetabular facet is shallowly concave that forms a broad dorsoventrally convex surface that smoothly connects the posteroventral surface of the iliac blade to the posterior internal surface of the iliac acetabular facet. The surface of the iliac acetabular facet of NMQR 3939 and NMQR 3351 is deep and smooth and contributes to approximately half of the dorsal surface acetabulum (Figs. 10C and 10D). The surface of the iliac acetabular facet faces ventrally due to the supraacetabular buttress, but has an extensive ventrolaterally facing surface as well. A slight ridge on the posterior end of the iliac acetabular facet that runs towards the centre of the acetabulum is present in NMQR 3351 and NMQR 3939. A similar ridge has been described for an isolated ilium attributed to a therocephalian from Antarctica (*Sidor et al., 2014*). A shallow depression is present posterior to this ridge in NMQR 3351, but the corresponding area in NMQR 3939 is not preserved (Figs. 10C and 10D).

The medial surface of the iliac blade of SAM-PK-K10698 (Figs. 10E and 10F) is extremely similar to that of *Regisaurus* described by *Kemp (1978)*. The medial surface is relatively flat, apart from the presence of a shallow triangular fossa, which is bordered by the broad ridges of the anteroventral and anterodorsal processes as on the lateral surface of the blade. The triangular depression terminates close to the base of the acetabular neck in the form of a deep crescent shaped depression, which represents the attachment point for the first sacral rib. Posterior to the ridge of the anterodorsal process at the centre of the ilium a second deep depression is present, which represents the attachment point for the second sacral rib. Posterior to this depression, on the posterior half of the iliac blade a small oval depression is present, which represents the attachment point for the third sacral rib. The medial surface of the iliac acetabular facet is slightly anteroposteriorly convex and dorsoventrally straight.

## Pubis

Pubes are preserved in NMQR 48, NMQR 3939, and NMQR 3568. NMQR 48 preserves the complete right pubis in dorsal view, but a bone, possibly a sacral rib, is overlying the

posterolateral margin of the blade (Fig. 11A). NMQR 3568 preserves the complete right pubis as a separate element (Figs. 11B–11E). NMQR 3939 only preserves the acetabular contribution of the left pubis (Figs. 10D, 11F and 11G). Measurements of the pubes are given in Table 5.

The pubis is comprised of a robust pubic head and a dorsoventrally thin pubic blade that are connected by a short anteroposteriorly constricted pubic neck. The pubic blade is mediolaterally and posteriorly expanded with a convex dorsal surface and concave ventral surface as described for *Olivierosuchus* (*Fourie & Rubidge, 2007*). Both the pubic blades of NMQR 48 and NMQR 3568 possess a strongly concave anterior margin, which forms a dorsoventrally thickened tuberosity on the anteromedial margin (Figs. 11A–11C and 11E). The pubic blade extends posteriorly with a smoothly convex medial margin. In NMQR 3568 the pubic blade extends laterally at the posterior end of the pubic blade attenuating to a rounded point (Figs. 11B and 11C). The medial margin of the pubic blade of NMQR 48 is largely covered by the aforementioned bone and is damaged in NMQR 3568, but it can be seen that the medial margin is deeply concave and recurves posteriorly at the pubic neck forming the anterior boarder of the obturator foramen as in other eutherocephalians, but differs from early-diverging therocephalians, which exhibit a pubic foramen that is entirely bounded in the pubis (*Boonstra, 1964*; *Fourie & Rubidge, 2009*).

Both the pubic blades of NMQR 48 and NMQR 3568 constrict to form the short pubic neck. The pubic neck expands anteroposteriorly and dorsoventrally to form the robust pubic head. The posterior surface of the pubic neck and pubic head of NMQR 48 and NMQR 3568 are shallowly depressed and are delimited from the dorsal surface by a posteroventrally sloped surface. In NMQR 3568 an obturator ridge extends from the ventral margin of the pubic head and runs along the anterior region of the ventral surface of the pubic blade terminating at the anteromedial tuberosity (Fig. 11C). The acetabular facet of NMQR 48 is not well exposed, but it can be seen that the ventral surface of the pubic head is anteroposteriorly expanded (Fig. 11A). The pubic acetabular facet of NMQR 3568 is well preserved (Figs. 11B–11D) and is oval in outline and bears a convex articulatory surface similar to that of *Scaloposaurus* (*Kemp, 1986*). In contrast, the pubic acetabular facet of NMQR 3939 is almost triangular and slightly concave, but this difference could be attributed to deformation and weathering along its ventrolateral margin (Figs. 10D, 11F and 11G).

## Ischium

Ischia are preserved in NMQR 48, NMQR 3939, and NMQR 3351. NMQR 3939 preserves an almost complete left ischium that is still articulated to the acetabulum, but the ischial blade has suffered considerable damage resulting in the loss of most of the medial and posterior margins (Figs. 10D, 11F and 11G). NMQR 3351 preserves the left ischium, but only the ischial acetabular facet and a portion of the posterior surface is exposed (Fig. 11H). NMQR 48 preserves a mostly complete left ischium that is damaged along its posterior margin (Fig. 11I). Measurements of the ischia are given in Table 5.

The ischium comprises a robust ischial head followed by a short and anteroposteriorly constricted ischial neck, which expands to form the dorsoventrally thin ischial blade. The

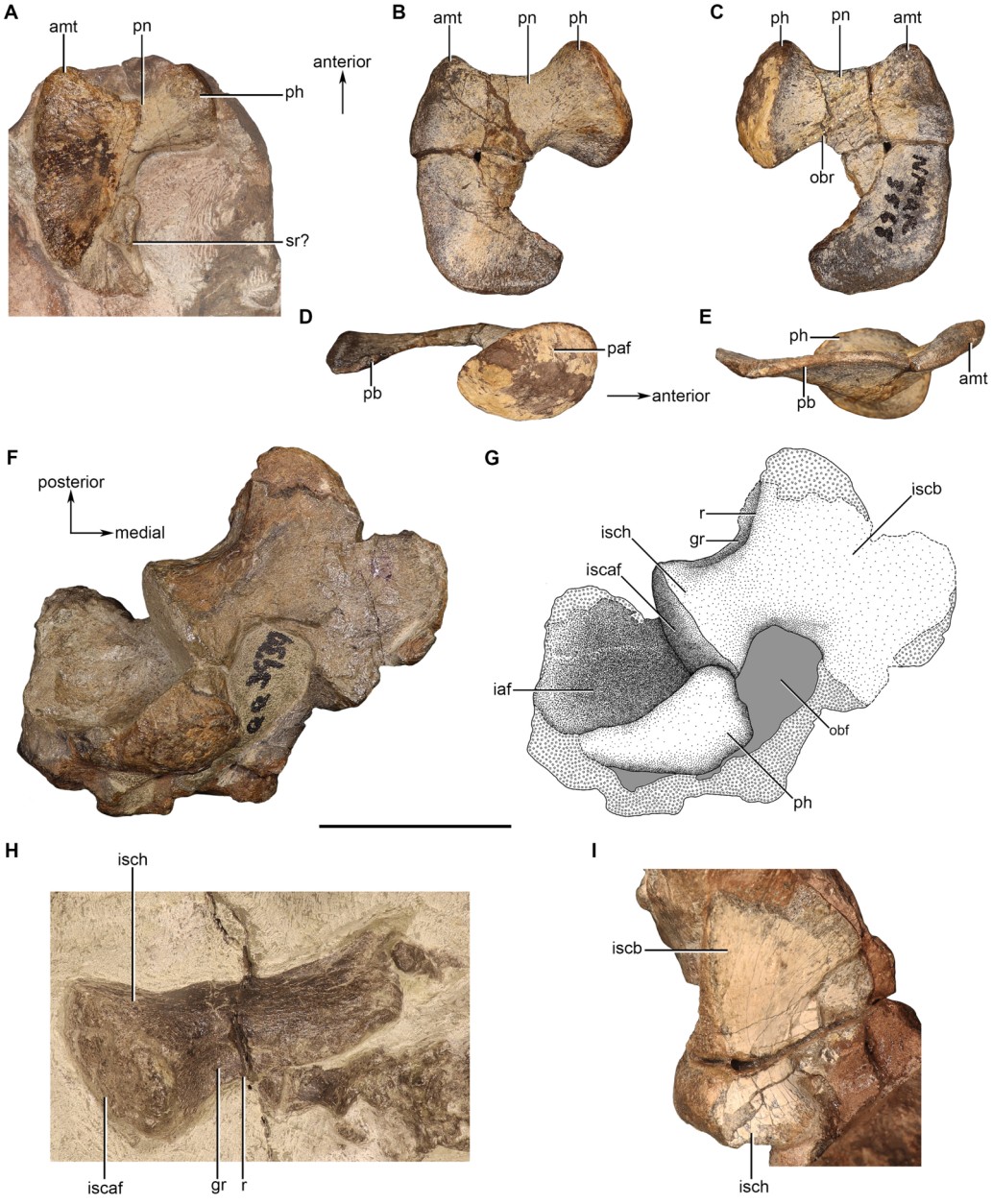

**Figure 11 Pubes and ischia of *Moschorhinus kitchingi*.** Right pubis of NMQR 48 (semi-transparent background) in dorsal view (A), right pubis of NMQR 3568 in dorsal (B), ventral (C), lateral (D), and medial (E) views, photograph (F) and stipple drawing (G) of the left pelvic girdle of NMQR 3939 in ventral view. Grey colouration indicates matrix and hatching indicates damaged surfaces. Left ischium of NMQR 3351 in posterior view (H), and right ischium of NMQR 48 in posteroventral view (I). Scale bar equals 50 mm. Abbreviations: amt, anteromedial tuberosity of the pubis; gr, groove; iaf, acetabular facet of the ilium; iscaf, acetabular facet of the ischium; isch, head of the ischium; iscb, blade of the ischium; obf, obturator foramen; obr, obturator ridge; paf, acetabular facet of the pubis; pb, blade of the pubis; ph, head of the pubis; pn, neck of the pubis; r, ridge; sr?, sacral rib. Photographs and illustration by Brandon P. Stuart.

preservation of both elements in NMQR 48 (Figs. 11A and 11I) allows for the comparison of overall size where it can be seen that the ischium is considerably larger than the pubis as reported for *Ericiolacerta* (*Watson, 1931*), *Scaloposaurus* (*Kemp, 1986*), and *Olivierosuchus* (*Fourie & Rubidge, 2007*). Despite being incomplete, the iliac blade of NMQR 3939 and NMQR 48 is large, and the ventral surface is flat as described for *Scaloposaurus* (*Kemp, 1986*), but differs from the convex dorsal surface described for *Olivierosuchus* (*Fourie & Rubidge, 2007*). The anterior margin of the iliac blade of NMQR 3939 is deeply concave and forms the posterior boarder of the obturator foramen (Figs. 11F and 11G). Although no specimen preserves a complete and articulated pubis and ischium, the pubes of NMQR 48 and NMQR 3568 along with the ischium of NMQR 3939 suggest that the obturator foramen was considerably large as described for *Choerosaurus* (*Haughton, 1929*), *Ericiolacerta* (*Watson, 1931*) and *Scaloposaurus* (*Kemp, 1986*). The posterior margin of the ischium of NMQR 3351 and NMQR 3939 is strongly concave and becomes dorsoventrally thickened along the ischial neck to form the robust anteroposteriorly and dorsoventrally expanded ischial head (Figs. 10D, 11F, 11G, 11H and 11I). A prominent triangular groove is present on the posterior surface of the ischium of NMQR 3939 and NMQR 3351 and is bordered by a sharp ridge that delimits the posterior and ventral surfaces (Figs. 10D and 11F–11H). The acetabular facet of NMQR 3939 and NMQR 3351 (Figs. 10D and 11F–11H) is larger than that of the pubis and is flat as described for *Scaloposaurus* (*Kemp, 1986*), but differs from the concave ischial acetabular facet described for *Olivierosuchus* (*Fourie & Rubidge, 2007*).

## Femur

Femora are preserved in NMQR 48, NMQR 3939, and NMQR 3351. NMQR 48 only preserves the proximal end of the left femur, but it is damaged. NMQR 3351 preserves both femora (Figs. 12A–12E). The right femur is complete, but it has a cross-sectional break through the proximal portion of the shaft (Figs. 12A and 12B). The proximal head is still articulated with the acetabulum, which only allows for the lateral, and partial anterior and posterior views to be observed. The left femur is incomplete and is represented by the proximal head and the proximal portion of the shaft, which are exposed in anterior, medial, and posterior views (Figs. 12C and 12D). NMQR 3939 preserves the entire left femur as three separate pieces; the proximal end, the proximal portion of the shaft, which is represented by a cast, and the distal shaft and distal end (Figs. 12F–12K).

The femur is a long, robust bone with slightly expanded proximal and distal ends that are connected by a well-defined diaphysis. The complete humerus and femur of NMQR 3351 allows for a comparison of both elements of which the femur is approximately 14% longer (Tables 3 and 6). In lateral and medial view, the femur of NMQR 3939 is only slightly sigmoidally curved, almost straight (Figs. 12H and 12I) similar to that of *Olivierosuchus* (*Fourie & Rubidge, 2007*) and other therocephalians in which the femur is known (*e.g.*, *Kemp, 1986*; *King, 1996*), but differing from the femur of *Regisaurus* (*Kemp, 1978*), which is markedly sigmoidally curved. The shaft of the femur of NMQR 3939 is greatly twisted so that the distal condyles form an acute angle of approximately 70° relative to the proximal head, differing from those of early-diverging therocephalians, which show

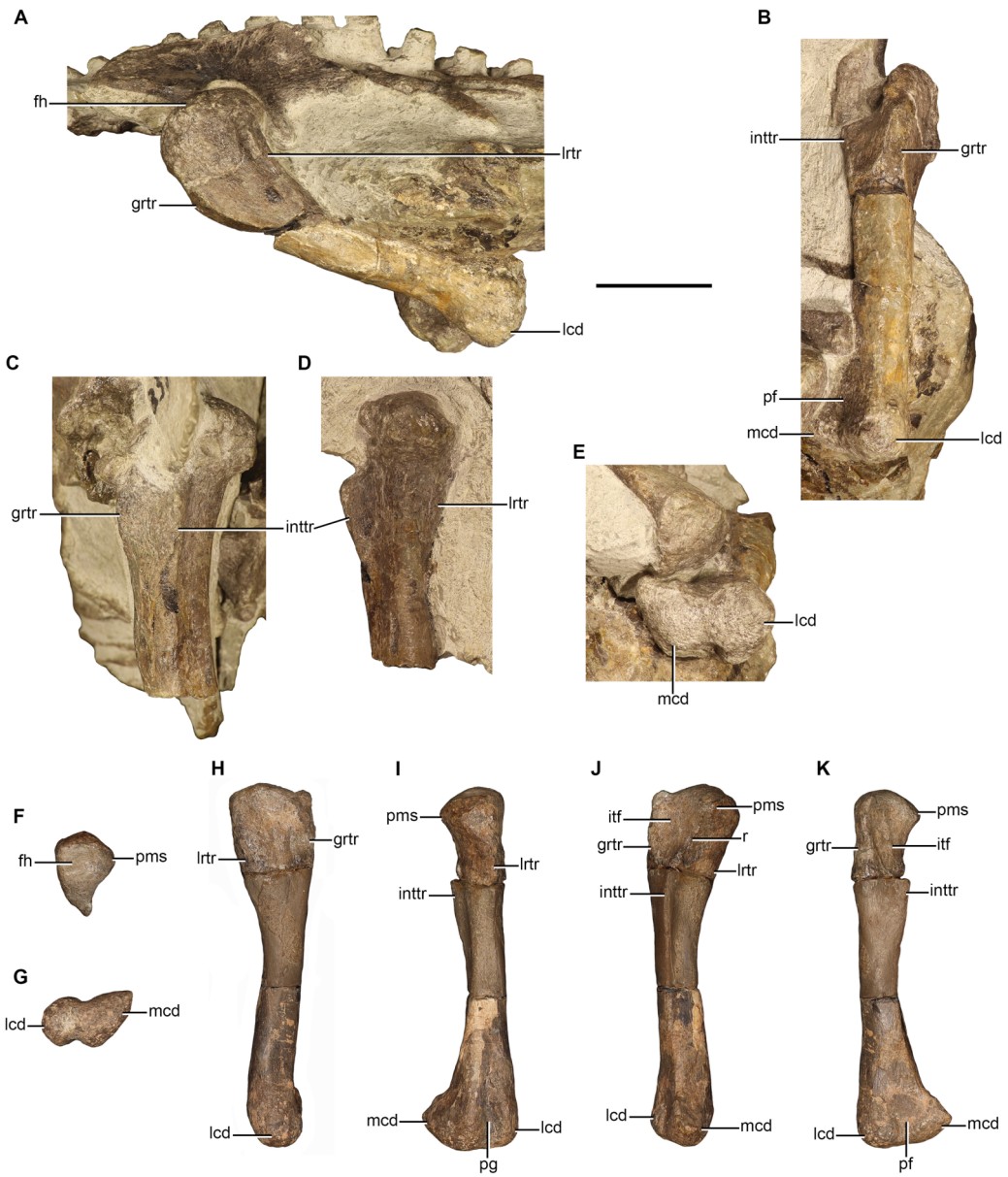

**Figure 12 Femora of *Moschorhinus kitchingi*.** Right femur of NMQR 3351 in lateral (A), posterior (B), and distal (E) view, left femur of NMQR 3351 in medial (C) and anterior (D) view, left femur of NMQR 3939 in proximal (F), distal (G), lateral (H), anterior (I), medial (J), and posterior (K). Scale bar equals 50 mm. Abbreviations: fh, femoral head; fos, fossa; grtr, greater trochanter of the femur ; itf, intertrochanteric fossa of the femur; lcd, lateral condyle; lrtr, lesser trochanter of the femur; mcd, medial condyle; pf, popliteal fossa; pg, patella groove; pms, proximomedial swelling of the femoral head. Photographs by Brandon P. Stuart.       

variable degrees of twisting between 30°–40° (*Boonstra, 1964*) and *Regisaurus*, which shows and angle of 40° (*Kemp, 1978*). The proximal surface of the femoral head of NMQR 3939 is convex anteroposteriorly and mediolaterally (Fig. 12F) as described for *Scaloposaurus* (*Kemp, 1978*), but differs from the flat proximal surface of the early-diverging therocephalian described by *Fourie & Rubidge (2009)*. The outline of the
proximal surface of the femoral head of NMQR 3939 is semi-circular anteriorly and becomes mediolaterally reduced, tapering posteriorly (Fig. 12F). A pronounced swelling is present on the proximomedial margin (Figs. 12F and 12I–12K) as in *Choerosaurus* (*Haughton, 1929*). The femoral head of both femora of NMQR 3351 exhibit a conspicuous anterior expansion of the proximoanterior margin (Figs. 12A and 12D), which is not observed in the femur of NMQR 3939, although the anterior surface is weathered (Fig. 12I).

The femora of NMQR 3351 and NMQR 3939 possess three trochanters that are positioned distally to the femoral head (Figs. 12A–12D, 12H and 12I–12K) as in all other therocephalians (*e.g.*, *Boonstra, 1964*; *Kemp, 1978*, *1986*; *Fourie & Rubidge, 2007*, *2009*). The tapering posterior end of the proximal surface of the femoral head of NMQR 3351 and NMQR 3939 forms the prominent greater trochanter, which presents as a sharp ridge in NMQR 3939 and a mediolaterally broad ridge in NMQR 3351 (Figs. 12A–12C, 12H, 12J and 12K). Damage to the posterior surface of the femoral head of NMQR 3939 has led to an unnatural protruded point (Figs. 12H and 12J). The greater trochanter is mediolaterally thickened distally along the posterior margin, forming a dorsoventrally short ridge on the lateral surface of the proximal end of the femur (Figs. 12A and 12H). The lesser trochanter is present on the anterior margin of NMQR 3351 and NMQR 3939 (Figs. 12A, 12D, 12I and 12J). The lesser trochanter of NMQR 3351 and NMQR 3939 is positioned more distally from the femoral head and is in the form of a slight mediolateral swelling, but it is much less developed than the greater trochanter. As with the greater trochanter, swelling of the lesser trochanter forms a short dorsoventral ridge on the anterior margin of the lateral surface of the proximal end in NMQR 3351 and NMQR 3939 (Figs. 12A and 12J). The lateral surface of the proximal ends of NMQR 3351 and NMQR 3939 is flat and bounded by these ridges.

The femora of NMQR 3351 and NMQR 3939 possess a well-developed internal trochanter that is positioned distal to the femoral head and along the midline of the medial surface (Figs. 12B–12D and 12I–12K). The midline position of the internal trochanter is similar to that described for other therocephalians (*e.g.*, *Boonstra, 1964*; *Kemp, 1978*, *1986*), but differs from *Choerosaurus* where it is positioned almost on the posterior margin of the medial surface (*Haughton, 1929*). A low, weak ridge extends distally from the proximomedial swelling to the proximal end of the internal trochanter in NMQR 3939 (Fig. 12J). The intertrochanteric fossa is an anteroposteriorly concave, depressed surface posterior to this ridge (Fig. 12J). The internal trochanter of NMQR 3351 and NMQR 3939 is a strong, triangular flange in anterior and posterior view, that protrudes medially producing a prominent adductor ridge along the shaft (Figs. 12D and 12J). The internal trochanter gradually recedes distally along the shaft in the form of a low ridge. The medial surface anterior to the internal trochanter of NMQR 3351 and NMQR 3939 is deeply concave (Fig. 12J).

The shaft of NMQR 3351 and NMQR 3939 expands mediolaterally distally to form the well-developed lateral and medial condyles (Figs. 12B, 12E, 12G, 12I and 12K). The patellar groove of NMQR 3939 is a short and deep notch that separates the lateral and medial condyles on the dorsal surface of the distal end (Fig. 12I). The ventral surface of the

distal end of NMQR 3351 and NMQR 3939 bears a mediolaterally broad and triangular popliteal fossa that separates the lateral and medial condyles (Figs. 12B and 12K). The lateral and medial condyles bear convex distal surfaces and encompass the entire distal surface of the femur (Figs. 12E and 12G). The lateral condyle of NMQR 3351 and NMQR 3939 is smaller than the medial condyle and is circular in outline in distal view (Figs. 12E and 12G). The medial condyle of NMQR 3351 and NMQR 3939 extends medioventrally and narrows distally to terminate as a sharp point so that it is approximately triangular in outline in distal view (Figs. 12E and 12G).

### Tibia

Tibiae are only preserved in NMQR 3351 and NMQR 3939. NMQR 3351 preserves the complete right tibia, which is exposed in posterior view (Fig. 13A). Both tibiae are preserved in NMQR 3939 as separate elements (Figs. 13B–13E). The right tibia is incomplete, missing the distal end, and has suffered considerable anteroposterior compression at its proximal end. The right tibia is represented by two pieces: the proximal end and most of the shaft, and a small portion of the distal shaft, represented by a cast (Figs. 13B and 13C). The left tibia is complete and is represented by three pieces: The proximal end and a portion of the proximal shaft, the mid and distal shaft represented by a cast, and the distal end (Figs. 13D and 13E).

The tibia is a relatively long bone with a mediolaterally expanded proximal end and an anteroposteriorly flattened shaft. The complete left tibia of NMQR 3939 is approximately 79% the length of the femur (Table 6). The right tibia of NMQR 3351 is only slightly longer, being approximately 82% the length of the femur. The tibia of NMQR 3351 is medially bowed with a convex medial margin and concave lateral margin (Fig. 13A). The lateral margin is more straight than convex in the tibiae of NMQR 3939 (Figs. 13B–13E). The bowed tibia of NMQR 3351 differs from the straight tibia described for *Ericiolacerta* (*Watson, 1931*), an early-diverging therocephalian (*Fourie & Rubidge, 2009*), and *Promoschorhynchus* (*Huttenlocker, Sidor & Smith, 2011*). The medial end of the proximal surface of NMQR 3351 and NMQR 3939 is convex and slightly anteroposteriorly constricted whereas the lateral end is anteroposteriorly expanded, fairly flat, and slopes ventrolaterally (Figs. 13A–13E). The cnemial crest of the right tibia of NMQR 3939 is distorted and is not observable in NMQR 3351. The cnemial crest of the left tibia of NMQR 3939 is present on the anterior surface of the proximal end and is positioned closer to the lateral end than the medial end (Fig. 13E). It is moderately well developed and overhangs the anterior surface of the proximal end (Fig. 13E). A deep dorsoventral groove is present approximately midway on the shaft and is positioned close to the lateral margin (Figs. 13A, 13B and 13D) similar to that described for *Regisaurus* (*Kemp, 1978*) and *Olivierosuchus* (*Fourie & Rubidge, 2007*). A sharp oblique ridge is present on the anterior surface of NMQR 3939 (Figs. 13C and 13E), similar to that described for *Regisaurus* (*Kemp, 1978*) and *Promoschorhynchus* (*Huttenlocker, Sidor & Smith, 2011*). The ridge originates weakly, close to the lateral margin of the proximal end and extends distally to the medial margin (Figs. 13C and 13E).

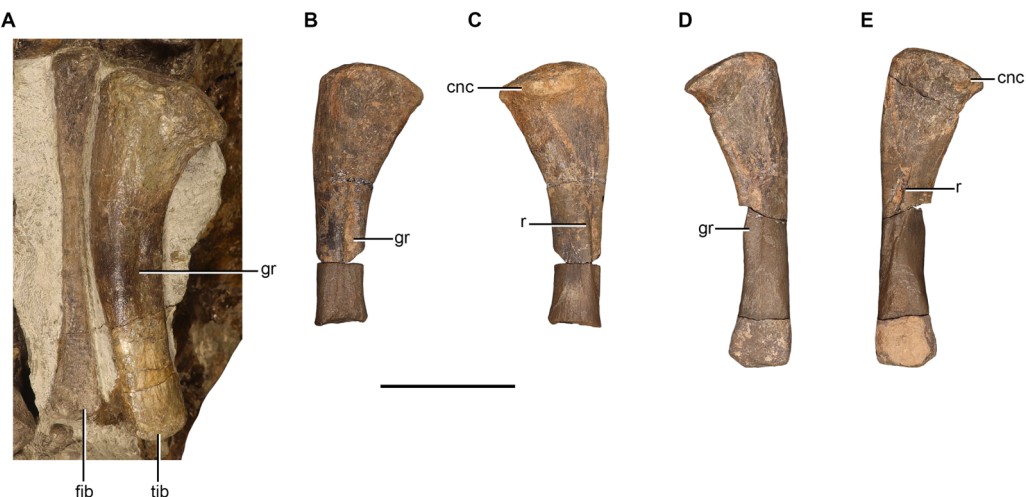

**Figure 13 Tibiae and fibula of *Moschorhinus kitchingi*.** Right tibia and fibula of NMQR 3351 in posterior view (A), right (B and C) and left (D and E) tibia of NMQR 3939 in posterior and anterior views. Scale bar equals 50 mm. Abbreviations: cnc, cnemial crest; fib, fibula; gr, groove; r, ridge; tib, tibia. Photographs by Brandon P. Stuart.

**Table 6 Measurements of the femora, tibiae, fibula of specimens of *Moschorhinus kitchingi*.**

| | NMQR 3939 | | NMQR 3351 | |
| --- | --- | --- | --- | --- |
| | **Left** | **Right** | **Left** | **Right** |
| **Femur** | | | | |
| Length | 145.07 | – | 111.98* | 178.70 |
| Length of proximal end | 33.99 | – | 48.17* | 48.28 |
| Width of proximal end | 24.84 | – | – | – |
| Width of distal end | 39.84 | – | – | 56.17 |
| **Tibia** | | | | |
| Length | 114.84 | 95.78* | – | 146.57 |
| Width of proximal end | 17.35 | 13.98* | – | 49.91 |
| Width of distal end | 13.30 | – | – | 23.65 |
| **Fibula** | | | | |
| Length | – | – | – | 145.31 |
| Width of proximal end | – | – | – | 19.46 |
| Width of distal end | – | – | – | 21.92 |

**Notes:**
All measurements are in mm.
* Indicates element is partially covered by matrix or incomplete.

## Fibula

NMQR 3351 is the only specimen that preserves a fibula (Fig. 13A). The fibula is a long, slender, and straight bone with slightly expanded proximal and distal ends that are connected by a constricted shaft. The fibula is a relatively featureless bone and similar to that described for other therocephalians (*e.g.*, *Attridge, 1956*; *Boonstra, 1964*; *Cys, 1967*;

*Kemp, 1978*, *1986*; *King, 1996*; *Fourie & Rubidge, 2007*, *2009*; *Huttenlocker, Sidor & Smith, 2011*).

## DISCUSSION

The excellent preservation and abundance of material described herein has provided one of the most detailed and comprehensive descriptions of a therocephalian postcranial skeleton to date. This has allowed us to present the first skeletal reconstruction of *Moschorhinus kitchingi* (Fig. 14) and provide a summary of the salient points of comparison that could be obtained from the current body of work of therocephalian postcranial anatomy. We opted not to include a phylogenetic analysis as *Moschorhinus* has already been scored for the few postcranial characters that are included in previous therocephalian matrices (*e.g.*, *Huttenlocker & Sidor, 2016*; *Liu & Abdala, 2022*). We also refrained from attempting to derive new postcranial characters for therocephalians as this is out of the scope of the current work, and moreover, is actively being worked on by the current authors and will be included in a future contribution.

### Therocephalian postcranial anatomy
#### Axial skeleton

The most complete axial skeletons known for therocephalians are that of *Mirotenthes* (*Attridge, 1956*), *Theriognathus* (*Brink, 1958b*), an early-diverging therocephalian (*Cys, 1967*), *Scaloposaurus* (*Kemp, 1986*), *Olivierosuchus* (*Fourie & Rubidge, 2007*), and a scylacosaurid therocephalian (*Fourie & Rubidge, 2009*). NMQR 3351 provides a total presacral vertebral count of 27 for *Moschorhinus*, which is consistent with that of *Mirotenthes, Theriognathus, Scaloposaurus, Olivierosuchus* (BP/1/3973), and the early-diverging therocephalian described by *Cys (1967)*. This is surprising because although this represents a small sample size of Therocephalia, these specimens represent genera from most of the major subclades, potentially indicating a highly conserved number of presacral vertebrae across the clade. This contrasts with other therapsid clades that show variable numbers of presacral vertebrae (*Fröbisch & Reisz, 2011*).

The degree of differentiation of the therocephalian axial skeleton has been cursorily discussed by *Kemp (1986)* in his description of a small skeleton of *Scaloposaurus*, of which he commented on the similarities to cynodonts, presumably early cynodonts such as *Procynosuchus* (*Kemp, 1980*) and *Thrinaxodon* (*Jenkins, 1970*). *Kemp (1986)* described discrete vertebral regions for *Scaloposaurus*, primarily defining these regions by the change in the angle of the zygapophyses in the cervical, thoracic, and lumbar regions, but also by the presence of short and horizontal ribs in the lumbar region. He also commented on the presence of transitional vertebral structures between these regions. The axial skeleton of NMQR 3351 bears remarkable similarities to that of *Scaloposaurus* with regards to the degree of differentiation, but also by the morphological changes of vertebral and rib structures at the same transitional regions.

The structural changes of the seventh cervical vertebrae of NMQR 3351 indicating the transition to the dorsal region are the change in the projection of the neural spine from anterodorsal to more dorsal, the absence of the lateral ridge on the lateral margin of the

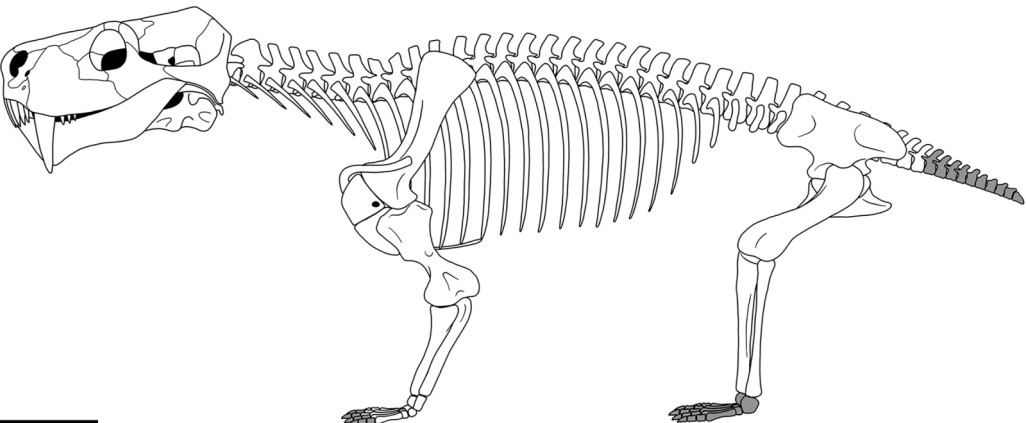

**Figure 14 Skeletal reconstruction of *Moschorhinus kitchingi*.** The skeletal proportions are based on the preserved elements of NMQR 3351. Grey indicate elements that are not preserved in any of the specimens. Scale bar equals 100 mm. Illustration by Brandon P. Stuart.

neural arch, the change in orientation of the postzygapophyses from horizontal to more vertical, and the change in the position of the transverse process from ventral to more dorsal, the orientation from ventrolateral to more lateral, and an increase in anteroposterior width. Similarly, the rib of the seventh cervical is markedly anteroposteriorly broader than the preceding cervical ribs and subequal in length to the succeeding dorsal ribs. Moreover, the first two dorsals are also morphologically intermediate from the preceding cervicals and succeeding mid-dorsals by bearing lower neural spines with posteroventrally and posterodorsally inclined dorsal margins respectively and by the presence of anteroposteriorly broad and laterally projected transverse processes. The transition from the mid-dorsals to the posterior dorsals in NMQR 3351 occurs from the 23rd to the 24th vertebra. This transition is indicated by the decrease in height and increase in mediolateral width as well as the increase in anteroposterior length at the base of the neural spines, the posterolateral orientation and broadening of the postzygapophyses, the shortening of the transverse processes anteroposteriorly and mediolaterally, and the shortening of the associated ribs.

### Pectoral girdle

The gross anatomy of the scapula of *Moschorhinus* is similar to other therocephalians by having an elongated scapular blade, a convex lateral surface, and by lacking an acromion process (*Attridge, 1956*; *Boonstra, 1964*; *Cys, 1967*; *Kemp, 1986*; *King, 1996*; *Fourie & Rubidge, 2007, 2009*; *Botha-Brink & Modesto, 2011*; *Huttenlocker, Sidor & Smith, 2011*; *Sigurdsen et al., 2012*). The scapula of *Moschorhinus* and other akidnognathid therocephalians is generally more robust than other eutherocephalian taxa, for example, the scapular blade of the similarly sized akidnognathids *Olivierosuchus* (*Botha-Brink & Modesto, 2011*) and *Promoschorhynchus* (*Huttenlocker, Sidor & Smith, 2011*) are anteroposteriorly wider compared to the extremely slender scapular blade of the whaitsioid *Mirotenthes* (*Attridge, 1956*). The scapulae of the smaller baurioid therocephalians are

slender and more similar to that of *Mirotenthes* (*Attridge, 1956*; *Kemp, 1986*; *King, 1996*; *Sigurdsen et al., 2012*). In contrast, the scapulae of early-diverging therocephalians has been described as being robust and dorsoventrally short (*Boonstra, 1964*; *Fourie & Rubidge, 2009*), but this is difficult to surmise at this point in time due to the limited descriptive work and taxonomic issues concerning these taxa.

At this point in time, very little differentiates the coracoid of *Moschorhinus* from that of early-diverging therocephalians and eutherocephalians. The procoracoid of *Moschorhinus* is similar to the procoracoids that have been described for early-diverging therocephalians and most other eutherocephalians by having the procoracoid foramen positioned solely within the procoracoid (*Broom, 1900*; *Attridge, 1956*; *Boonstra, 1964*; *Cys, 1967*; *Fourie & Rubidge, 2007*; *Botha-Brink & Modesto, 2011*; *Huttenlocker, Sidor & Smith, 2011*; *Sigurdsen et al., 2012*; *Fourie, 2013*). The position of the procoracoid foramen in therocephalians is considered to be phylogenetically important because in the deeply nested Triassic bauriamorph baurioids *Scaloposaurus* (*Kemp, 1986*) and *Ericiolacerta* (*Watson, 1931*) it is positioned on the procoracoid-coracoid suture and on the scapula-procoracoid suture in the derived bauriid *Bauria* (*Watson, 1931*). Much like the coracoid, the clavicle of *Moschorhinus* is very similar to that of other therocephalians, apart from the scalloped edges of the ventral end, which may represent a condition unique to the taxon.

The interclavicle of *Moschorhinus* is remarkably small in relation to the rest of the skeleton and the general shape is approximately similar to the interclavicles of other eutherocephalians, but variation of the structures on the anterior and posterior rami is evident among different eutherocephalian taxa. The interclavicle of *Moschorhinus* and indeed other eutherocephalians differs from the large spoon-shaped interclavicle of early-diverging therocephalians (*Boonstra, 1964*; *Fourie & Rubidge, 2009*). The circular sternum shape of *Moschorhinus* is consistent among different specimens and appears to be a morphology unique to the taxon. In contrast to the sternum seen in the akidnognathids *Promoschorhynchus* and *Olivierosuchus* which are oval in shape (*Fourie & Rubidge, 2007*; *Botha-Brink & Modesto, 2011*) and diamond-shaped in the baurioids *Ictidosuchoides*, *Tetracynodon darti* and *Scaloposaurus* (*Kemp, 1986*; *Sigurdsen et al., 2012*; *Fourie, 2013*). Aside from sternal shape, the crenulations on the lateral margins, presence of a posterior notch, and the condition of the ventromedial ridge is highly variable among eutherocephalian taxa. The weak crenulations on the sterna of *Moschorhinus* are similar to most other eutherocephalian taxa apart from the deeply-crenulated sternum of *Tetracynodon darti* (*Sigurdsen et al., 2012*). None of the sterna of *Moschorhinus* have a posterior notch similar to the akidnognathid *Promoschorhynchus* (*Huttenlocker, Sidor & Smith, 2011*). A posterior notch is however present in the akidnognathid *Olivierosuchus* (*Botha-Brink & Modesto, 2011*) and the baurioids *Tetracynodon darti* (*Sigurdsen et al., 2012*) and *Scaloposaurus* (*Kemp, 1986*). The condition of the ventromedial ridge of the three sterna of *Moschorhinus* described herein is interesting because it indicates that the anterior extension of this ridge increases with an increase in skull size, which suggests the development of this structure is associated with ontogeny as suggested by *Sigurdsen et al. (2012)*. In other eutherocephalian taxa, which are inferred to represent ontogenetically
mature individuals, the ventromedial ridge extends to the centre of the sternum (*e.g.*, *Olivierosuchus* and *Promoschorhynchus*), similar to the condition seen the in the largest specimen of *Moschorhinus* (NMQR 3351). However, a well-developed ventromedial ridge that extends along the length of the entire sternum is present in *Tetracynodon darti* and may represent a condition unique to that taxon.

### Forelimb

The humerus of *Moschorhinus* is only similar to that of early-diverging therocephalians in terms of the degree of its robustness and expansion of the epiphyses (*Boonstra, 1964*; *Cys, 1967*), contrasting with the gracile nature of other smaller eutherocephalian taxa, which generally have comparatively narrower epiphyses (*Attridge, 1956*; *Kemp, 1986*; *Sigurdsen et al., 2012*). Nevertheless, the humerus of *Moschorhinus* is differentiated from that of early-diverging therocephalians by the presence of a large and well-developed deltopectoral crest and the absence of an ectepicondylar foramen (*Boonstra, 1964*; *Fourie & Rubidge, 2009*). Overall, the radius and ulna of *Moschorhinus* is very similar to those described for other therocephalians.

The manus of *Moschorhinus* is short and broad, more closely resembling those described for early-diverging therocephalians (*Boonstra, 1964*; *Cys, 1967*; *Fontanarossa et al., 2018*), rather than the akidnognathid *Olivierosuchus*, which has a comparatively more slender manus (*Fourie & Rubidge, 2007*; *Botha-Brink & Modesto, 2011*) and baurioid eutherocephalians (*e.g.*, *Tetracynodon darti, Scaloposaurus, Ericiolacerta*), which generally have a longer and an even more slender manus (*Watson, 1931*; *Kemp, 1986*; *Fontanarossa et al., 2018*). The quadrangular radiale of *Moschorhinus* is similar to the radiale described for early-diverging therocephalians (*Kümmel et al., 2020*) and most eutherocephalians (*Fourie & Rubidge, 2007*; *Botha-Brink & Modesto, 2011*; *Kümmel et al., 2020*), but differs from the square radiale of the baurioid *Tetracynodon darti* and the rectangular radiale of the derived bauriid *Microgomphodon* (*Fontanarossa et al., 2018*; *Kümmel et al., 2020*). The ulnare of eutherocephalians is generally rectangular in shape, a condition that is also seen in *Moschorhinus*, but contrasts with the distinctly hourglass-shaped ulnare of early-diverging therocephalians (*Cys, 1967*; *Fourie & Rubidge, 2009*; *Kümmel et al., 2020*) and the early-diverging baurioid *Ictidosuchoides* (*Fourie, 2013*; *Kümmel et al., 2020*). The fusion of distal carpals IV and V observed in *Moschorhinus* represents the common condition in therocephalians, but differs from the separate ossifications seen in the early-diverging baurioid *Ictidosuchoides* and the inferred cartilaginous distal carpal V of *Tetracynodon darti* (*Fontanarossa et al., 2018*; *Kümmel et al., 2020*).

### Pelvic girdle

The anatomy of the ilium of *Moschorhinus* is consistent with that of other therocephalians with the presence of two processes on the anterior margin of the blade. These processes are less distinctive in early-diverging therocephalians, which appears to be the case in *Moschorhinus* as well, contrasting with the delicate finger-like process observed in the akidnognathid *Olivierosuchus* (*Fourie & Rubidge, 2007*) and the baurioids *Regisaurus* (*Kemp, 1978*) and *Scaloposaurus* (*Kemp, 1986*). The articulated pelvic girdle of NMQR

3939 shows that the blade of the ischium extends more horizontally, rather than vertically, which is consistent with the condition observed in early-diverging therocephalians and eutherocephalians where the pubis and the ischium form a horizontally orientated puboischiatic plate (*Boonstra, 1964*; *Cys, 1967*; *Kemp, 1986*; *Fourie & Rubidge, 2007*, *2009*). At present, very little differentiates the pubis and ischium of *Moschorhinus* from that of other eutherocephalians, but they do contrast with the pubis and ischium of early-diverging therocephalians, which have a pubic foramen rather than the obturator foramen, which is present in eutherocephalians.

### Hind limb

The femur of *Moschorhinus* conforms to the general structure of other therocephalians with the presence of three distinct trochanters and a slight curvature (*Boonstra, 1964*; *Kemp, 1978*, *1986*; *King, 1996*; *Fourie & Rubidge, 2007*, *2009*). The midline position of the internal trochanter of *Moschorhinus* is similar to that described for other therocephalians, apart from the baurioid *Choerosaurus*, which has the internal trochanter positioned more posteriorly (*Haughton, 1929*). The degree of curvature of the femur of *Moschorhinus* is similar to that seen in the akidnognathid *Olivierosuchus* (*Fourie & Rubidge, 2007*), but contrasts with the strongly sigmoidally curved femur of the baurioid *Regisaurus* (*Kemp, 1978*). The medially bowing is the only feature that differentiates the tibia of *Moschorhinus* from the tibia of the akidnognathids *Olivierosuchus* and *Promoschorhynchus* (*Fourie & Rubidge, 2007*; *Huttenlocker, Sidor & Smith, 2011*). No notable differences between the fibula of *Moschorhinus* and the fibula of other therocephalians can be observed at this time.

## Body size and mass of NMQR 3351 and palaeobiological insights

Recent records from China have shed new light on the early diversification of the akidnognathids showing that the Laurasian taxa were primarily medium-to-large sized (BSL > 200 mm) with *Jiufengia* representing the largest Laurasian akidnognathid taxon at a BSL of 250 mm (*Liu & Abdala, 2019*). In the Karoo Basin, only *Moschorhinus* (and the stratigraphically lower occurring whaitsiid *Theriognathus*) attained a large body size similar to that of early-diverging middle Permian therocephalians (*Huttenlocker & Abdala, 2015*; *Kammerer & Masyutin, 2018a*). Crania of *Moschorhinus* ranges from 136 to 262 mm (*Huttenlocker & Botha-Brink, 2013*) with NMQR 3351 at a BSL of 240 mm being the largest individual to preserve a complete skeleton. Based on the dimensions of the skeletal material of NMQR 3351, we estimate a minimum body length of approximately 1.1 m with a maximum length likely not exceeding 1.3 m (Fig. 14). Additionally, the preservation of both the humerus and femur of NMQR 3351 allows for the estimation of the body mass of NMQR 3351 using the minimum circumference of the shafts of these elements and the general equation for quadrupedal tetrapods from *Campione & Evans (2012)*. Using this general formula (see Supplemental Information), we estimate a body mass of approximately 84.31 kg for NMQR 3351, which is comparable in size to a large male leopard (*Panthera pardus*) (*Stuart & Stuart, 2017*).

Gorgonopsians were the primary components of the terrestrial carnivore guild in the Karoo Basin during the late Permian (Lopingian). However, the majority of the generic

diversity and abundance is lost leading up to the *Daptocephalus* AZ (Wuchiapingian–Changhsingian) with the disappearance of the large-bodied rubidgeines *Rubidgea* and *Aelurognathus* within the older *Dicynodon-Theriognathus* Subzone and the smaller-bodied taxa (*e.g.*, *Cyonosaurus*, *Arctognathus*, and *Lycaenops*) within the younger *Lystrosaurus maccaigi-Moschorhinus* Subzone (*Kammerer, 2015*; *Viglietti, 2020*; *Viglietti et al., 2021*; *Kammerer et al., 2023*; *Benoit et al., 2024*). Recently, *Kammerer et al. (2023)* showed that a step-wise and rapid turnover occurred in the apex predatory niche in the Karoo Basin with the last gorgonopsian to occupy the role being the newly described Luarasian immigrant taxon *Inostrancevia africana*. Although *I. africana* is undoubtably the largest therapsid predator to occur in the Karoo Basin during the *Lystrosaurus maccaigi-Moschorhinus* Subzone, it should be noted that a considerable diversity of small-to-medium sized theriodonts: gorgonopsians (*e.g.*, *Arctognathus* and *Cyonosaurus*), the cynodont *Vetusodon*, and the akidnognathid therocephalian *Promoschorhynchus* are present along with *Moschorhinus* in this faunal assemblage (*Abdala et al., 2019*; *Viglietti, 2020*; *Viglietti et al., 2021*; *Kammerer et al., 2023*; *Benoit et al., 2024*). This is interesting to highlight because, as has been noted for the largest gorgonopsians (*Kammerer, 2016*), the presence of several similarly sized taxa implies that some mechanisms of niche partitioning must have occurred in this 'mesocarnivore' type of guild.

The postcranial anatomy of gorgonopsians, like therocephalians, is generally poorly understood, but a recent resurgence of descriptive work has provided new data points on this clade (*e.g.*, *Kammerer & Masyutin, 2018b*; *Sidor, 2022*; *Bendel et al., 2022*, *2023*; *Sidor & Mann, 2024*). This work, with the addition of the pioneering monograph on *Lycaenops* by *Colbert (1948)*, provides a relatively good idea of the general bauplan of the small-bodied non-rubidgeine gorgonopsians, and thus offers a relevant point of comparison. The relative proportions of the postcranial elements of *Moschorhinus* shows that the appendicular skeleton, particularly the scapula, humerus, and femur are remarkably robust indicating a powerful and stocky bauplan (Fig. 14) in comparison to the gracile nature of small-bodied gorgonopsians (*Colbert, 1948*; *Kammerer & Masyutin, 2018b*; *Bendel et al., 2023*). Interestingly, the manual elements (metacarpals and phalanges) of *Moschorhinus* are surprisingly short compared to the stylopod and zeugopod (Fig. 14) in contrast to that of gorgonopsians which are generally longer and larger (*Bendel et al., 2023*). The general morphotypic variation of the manus, as noted by *Bendel et al. (2023)*, coupled with the more robust appendicular anatomy of *Moschorhinus* may reflect a difference in lifestyle and ecology between *Moschorhinus* and contemporaneous gorgonopsians. Additionally, the short and robust skull, and the unstriated, enlarged, and conical canines of *Moschorhinus* have been interpreted as an adaptation for withstanding sustained periods of being imbedded in prey (*van Valkenburg & Jenkins, 2002*), contrasting with the serrated and blade-like canines of gorgonopsians, which were better suited for slashing as well as a 'puncture and pull' type technique (*Kammerer, 2016*; *Whitney et al., 2020*). The combination of these cranial and postcranial traits, with the addition of relatively rapid and consistent growth rates (*Huttenlocker & Botha-Brink, 2013*; *Huttenlocker, 2014*), may have provided a competitive advantage for *Moschorhinus* over similarly sized gorgonopsians. Furthermore, resource scarcity across the extinction acme

likely disadvantaged larger predators like *I. africana* as opposed to the smaller-bodied *Moschorhinus*, which ultimately allowed for a successful, but brief occupation of the apex predator role during the PTME.

## CONCLUSIONS

For the first time, the postcranial anatomy of the large predatory eutherocephalian *Moschorhinus kitchingi* is described. The axial skeleton of *Moschorhinus* is remarkably similar to what is known for other therocephalians, particularly in terms of the presacral count and differentiation of the axial column. However, as many taxa are not known from complete axial skeletons, potential morphological variation, and varying degrees of differentiation among temporally or phylogenetically distant taxa may be present and will likely prove to be a productive avenue of research. The appendicular skeleton of *Moschorhinus* is only similar to that of early-diverging therocephalians in terms of robustness, but is otherwise differentiated by the presence of a small interclavicle and an ossified sternum, the absence of an ectepicondylar foramen on the humerus, and the presence of an obturator foramen bound between the pubis and ischium rather than a pubic foramen, all of which are postcranial characters that are shared by all eutherocephalians. The most notable morphological variation between eutherocephalians is in the pectoral girdle, particularly the scapula, scapulocoracoid foramen, interclavicle, and sternum, as well as the manus. Despite the novel data presented here, the extent of morphological variation and disparity of the therocephalian skeleton is not adequately constrained as most taxa are only known from cranial material; few are known from well-preserved and articulated postcranial skeletons. As such, the intergeneric variation of therocephalian postcrania will only be fully understood following the description of new specimens representing a broad range of therocephalian taxa and postcranial comparisons. This work provides a step in that direction.

## INSTITUTIONAL ABBREVIATIONS

**BP**          Evolutionary Studies Institute (formerly the Bernard Price Institute for Palaeontological Research), University of the Witwatersrand, Johannesburg, South Africa.

**CGS**         Council for Geosciences, Pretoria, South Africa.

**NHMUK**       Natural History Museum, London, United Kingdom.

**NMQR**        National Museum, Bloemfontein, South Africa.

**SAMI Iziko**  South African Museum, Cape Town, South Africa.

## ACKNOWLEDGEMENTS

For access to specimens in their care we thank E. Butler (National Museum, Bloemfontein), B. Zipfel and S. Jirah (Evolutionary Studies Institute, University of the Witwatersrand, Johannesburg), N. Mchunu and Z. Sibiya (Council for Geosciences, Pretoria), Z. Skosan and C. Browning (Iziko, South African Museum, Cape Town). We thank the fossil preparators S. Ledibane, S. Chaka, T. Ntsala at the National Museum,

Bloemfontein for the preparation of material. Finally, we thank B. Peecook and an anonymous reviewer for their helpful suggestions on improving the manuscript.

### Funding

Brandon P. Stuart was supported by the Palaeontological Scientific Trust (PAST) and Genus: DST-NRF Centre of Excellence in Palaeosciences. Jennifer Botha was supported by the National Research Foundation (UID 117704), PAST and GENUS: DST-NRF Centre of Excellence in Palaeosciences. Adam K. Huttenlocker has been funded by the National Science Foundation (award NSF-DEB 2325381). The funders had no role in study design, data collection and analysis, decision to publish, or preparation of the manuscript.

### Grant Disclosures

The following grant information was disclosed by the authors:
Palaeontological Scientific Trust (PAST).
Genus: DST-NRF Centre of Excellence in Palaeosciences.
National Research Foundation: UID 117704.
PAST and GENUS: DST-NRF Centre of Excellence in Palaeosciences.
National Science Foundation: NSF-DEB 2325381.

### Competing Interests

The authors declare that they have no competing interests.

### Author Contributions

- Brandon P. Stuart conceived and designed the experiments, performed the experiments, analyzed the data, prepared figures and/or tables, authored or reviewed drafts of the article, and approved the final draft.
- Adam K. Huttenlocker conceived and designed the experiments, analyzed the data, authored or reviewed drafts of the article, and approved the final draft.
- Jennifer Botha conceived and designed the experiments, analyzed the data, authored or reviewed drafts of the article, and approved the final draft.

### Data Availability

The specimens are reposited as follows:
- NMQR 3351 (Moschorhinus kitchingi)–National Museum, Bloemfontein, South Africa.
- NMQR 3939 (Moschorhinus kitchingi)–National Museum, Bloemfontein, South Africa.
- NMQR 3568 (Moschorhinus kitchingi)–National Museum, Bloemfontein, South Africa.
- NMQR 48 (Moschorhinus kitchingi)–National Museum, Bloemfontein, South Africa.

- CGS GHG299 (Moschorhinus kitchingi)–Council for Geosciences, Pretoria, South Africa.
- BP/1/4227 (Moschorhinus kitchingi)–Evolutionary Studies Institute, University of the Witwatersrand, Johannesburg, South Africa.
- SAM-PK-K10698 (Moschorhinus kitchingi)–Iziko South African Museum, Cape Town, South Africa.

## Supplemental Information

Supplemental information for this article can be found online at http://dx.doi.org/10.7717/peerj.17765#supplemental-information.

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
