# Peer review of "The postcranial anatomy of Moschorhinus kitchingi (Therapsida: Therocephalia) from the Karoo Basin of South Africa"

_PeerJ, doi:10.7717/peerj.17765_

## Round 0.1 · original submission · Minor Revisions

Please, take all the reviewers' comments into account and address them either in the text or in a separate answer.

·

Basic reporting

Stuart et al. provide an excellent and valuable postcranial description of the Permo-Triassic boundary crossing therocephalian Moschorhinus kitchingi. Therocephalian postcranial anatomy is poorly known generally, and the authors clearly figure a great deal. The skeletal is great.
The requirements of Basin Reporting at PeerJ are certainly met, though I will provide some minor edits and recommendations for changes below.

Experimental design

No comment. This manuscript meets the standards for PeerJ Experimental Design.

Validity of the findings

No comment. This manuscript meets the standards for PeerJ Validity of Findings.

Additional comments

Please see the following line edits/suggestions:

Keywords: "Therocephalia" rather "than therocephalians"
Line 88: Though you quantify some therocephalians later in text, I think here you should provide a range of BSL to help unfamiliar readers understand what "small-to-large-bodied" means.
Line 108: This topic sentence could end after "ecosystems" on line 109. Then the following sentence rewritten to clearly state the role gorgonopsians usually occupied in late Permian Karoo systems before their demise (unlike Moschorhinus).
Line 151: It would be easy to state where outside the Karoo Basin (Laurasian formations in Russia & China) other recent specimen descriptions come from.
Line 155: Could you provide an estimate of # of therocephalian genera vs. # with postcranial remains; or maybe remains described? It would drive home the fact that we do not have much in the literature on their anatomy.
Line 160: can you give us a brief insight into the range of body sizes (BSL? femur?) and state the stratigraphic levels?
Line 221: Could you add the middle initial of the senior James Kitching?
Line 228: "Changxingian" is an accepted alternate spelling of the terminal stage of the Permian, but perhaps consider using "Changhsingian".
Line 1384: Here, and throughout the remaining Discussion, "Laurasian" is misspelled as "Luarasian".
Lines 1402-1413: Though the evolving AZ scheme will continue to evolve, I think you can be more precise in how you talk about the raw distribution of 'theriodonts' in the two subzones of the Daptocephalus AZ as they were figured in the references cited; particualrly Viglietti. For instance: the DAZ is over three million years long. Aelurognathus disappears in the lower subzone, and Vetusodon appear at the end of the upper subzone. Some of the small gorgons make it well into the upper subzone alongside Moschorhinus, but don't overlap with Vetusodon like Moschorhinus does.

Reviewer 2 ·

Basic reporting

This manuscript represents a very valuable and novel contribution, as it, for the first time in decades, addresses a detailed description and discussion of the postcranial anatomy of therocephalian therapsids. I very much enjoyed the read, but must admit that parts of the description appeared somewhat lengthy and in parts even superfluous. Much detail is being provided on specimen numbers and preservation rather than on more detailed aspects of the actual general anatomy, This could be improved by tightenting up the text to provide a more concise and efficient description. Nonetheless and other than that I generally consider the manuscript close to being publishable as is. I only have very minor comments and suggestions for changes. This includes in particular the request for interpretive drawings of Figures 6 and 11m which are otherwise very difficult to depict from the simply labeled photographs. Further, "Laurasian" is misspelled (3 times) throughout the manuscript and should be corrected. All in all, I greatly appreciate the effort provided by the authors and look forward to seeing this study published.

Experimental design

no comment

Validity of the findings

no comment

---

## Round 0.2 · accepted · Accept

Thank you for addressing all reviewers' comments. I accept the current version and think that the manuscript is ready for publication. Congratulations to the authors.